# Probing the critical nucleus size in tetrahydrofuran clathrate hydrate formation using surface-anchored nanoparticles

Han Xue[1,6], Linhai Li[1,6], Yiqun Wang[2], Youhua Lu[1], Kai Cui[1], Zhiyuan He [1], Guoying Bai [1], Jie Liu[1,3] ✉, Xin Zhou [2,4] ✉ & Jianjun Wang [1,3,5] ✉

Controlling the formation of clathrate hydrates is crucial for advancing hydrate-based technologies. However, the microscopic mechanism underlying clathrate hydrate formation through nucleation remains poorly elucidated. Specifically, the critical nucleus, theorized as a pivotal step in nucleation, lacks empirical validation. Here, we report uniform nanoparticles, *e.g.*, graphene oxide (GO) nanosheets and gold or silver nanocubes with controlled sizes, as nanoprobes to experimentally measure the size of the critical nucleus of tetrahydrofuran (THF) clathrate hydrate formation. The capability of the nanoparticles in facilitating THF clathrate hydrate nucleation displays generally an abrupt change at a nanoparticle-size-determined specific supercooling. It is revealed that the free-energy barrier shows an abrupt change when the nanoparticles have an approximately the same size as that of the critical nucleus. Thus, it is inferred that THF clathrate hydrate nucleation involves the creation of a critical nucleus with its size being inversely proportional to the supercooling. By proving the existence and determining the supercooling-dependent size of the critical nucleus of the THF clathrate hydrates, this work brings insights in the microscopic pictures of the clathrate hydrate nucleation.

Clathrate hydrates, crystals in which the guest molecules are encapsulated in the hydrogen-bonded water nanocages[1,2], are showing broader utilizations in numerous fields[3,4], e.g., storing and transporting natural gas and hydrogen[5,6], sequestrating green-house gas[7,8]. In these technologies, controlled formation of the clathrate hydrates, involving the facilitating or inhibiting of the process, is the most dominant but challenging; therefore, fundamental insights into the microscopic mechanisms of clathrate hydrate formation are of paramount importance. In recent works, the formation of the clathrate hydrates is commonly described as a unified nucleation process of the first-order phase transitions, that is, the critical nucleus originates firstly from the local thermodynamic fluctuation, and afterwards the spontaneous growth of the clathrate nucleus occurs[9,10]. However, the detailed pictures at different length scales for the formation of clathrate hydrates are still lacked. Several hypotheses, such as the labile cluster hypothesis[11], local structuring hypothesis[12–15], and the blob model[16–19], were proposed as the initial step during the clathrate nucleation based on molecular simulations to elucidate the microscopic formation pathway of guest molecules encapsulated in water nanocages from aqueous solution. There is far less understanding of the further formation of clathrate hydrate through the nucleation process; particularly the information about the critical nucleus of clathrate is missing,

[1]Beijing National Laboratory for Molecular Science, Key Laboratory of Green Printing, Institute of Chemistry, Chinese Academy of Sciences, Beijing 100190, China. [2]School of Physical Sciences, University of Chinese Academy of Sciences, Beijing 100049, China. [3]School of Chemical Sciences, University of Chinese Academy of Sciences, Beijing 100049, China. [4]Wenzhou Institute, University of Chinese Academy of Sciences, Wenzhou 325001, China. [5]Technical Institute of Physics and Chemistry, Chinese Academy of Sciences, Beijing 100190, China. [6]These authors contributed equally: Han Xue, Linhai Li. ✉e-mail: liujie123@iccas.ac.cn; xzhou@ucas.ac.cn; wangjianjun@mail.ipc.ac.cn

which has been regarded as the control step of the formation of clathrate hydrates[20–28]. Due to the intrinsic feature of the critical nucleus, it is difficult to experimentally probe the small (nanometers), transient (nanoseconds), as well as randomly and rarely occurring critical nucleus. In a previous work[29], we introduced a methodology to investigate the critical nucleus size of ice by assessing the ice nucleation-promoting ability of nanoparticles with uniform sizes. It was expected to provide a general technique to investigate various phase transition nucleation processes[30].

In this work, we employ the technique to probe the critical nucleus size of the THF clathrate hydrate from its aqueous solution. Differing from the high-pressure requirement needed for gas clathrate hydrates, the capability of THF hydrates to form clathrates independently and as binary guest hydrates with gases (such as methane and hydrogen) under relatively mild experimental conditions makes them particularly intriguing for clathrate hydrate research[6,31]. Efforts have been put to study the nucleation process of THF hydrates, for example, via the x-ray Raman scattering measurements[14], and via regulating through antifreeze proteins (AFPs)[32]. The achieved conclusions were thought to be very helpful for the understanding of the gas clathrate hydrates. Here, we applied various size-uniform nanoparticles (involving graphene oxide nanosheets with size about 31 nm, 38 nm and 46 nm, gold or silver nanocubes with the size about 45 nm and 70 nm, respectively) and achieve highly-consistent results on the supercooling ($\Delta T$) · dependent of the critical nucleus size of THF hydrates.

We experimentally proved the formation of a critical nucleus as the key step of the phase transition of THF clathrate hydrate; and it is further showed that the spherical radius of the critical nucleus of the THF hydrates in nm is about $\frac{300\,°C}{\Delta T}$, which is about a few times larger than that of ice in experiment[29] and that of gas clathrate hydrates in simulations[19], but seems to be consistent with a recent (relevant) cryo-SEM experimental measurement, which reported the formation of about 10–30 nm nanoclusters in diameter of THF clathrate hydrates before the appearance of THF clathrate crystals at the supercooling $\Delta T = 20\,°C$[33]. The results indicate that the microscopic properties of the formed critical nuclei of THF hydrates and their surrounding solution differ from their macroscopic counterparts. This provides insights about the detailed picture of the THF nucleation for further investigation.

## Results

### Nucleation of THF clathrate on surfaces anchored nanosheets

We select THF sII clathrate hydrate (THF: water=1:17 in molar ratio) as a model[30,32,34–41] to experimentally investigate the nucleation process of clathrates. The clathrate hydrate nucleation was investigated via observing the influence of the size-controlled graphene oxide (GO) nanosheets on the freezing of deposited THF/water solution drops on supercooled substrate surfaces (illustrated in Fig. 1a and Supplementary Fig. 1), which was carried out in a sealed cell (the drop does not fill the cell completely) for avoiding liquid evaporation out of the cell (see

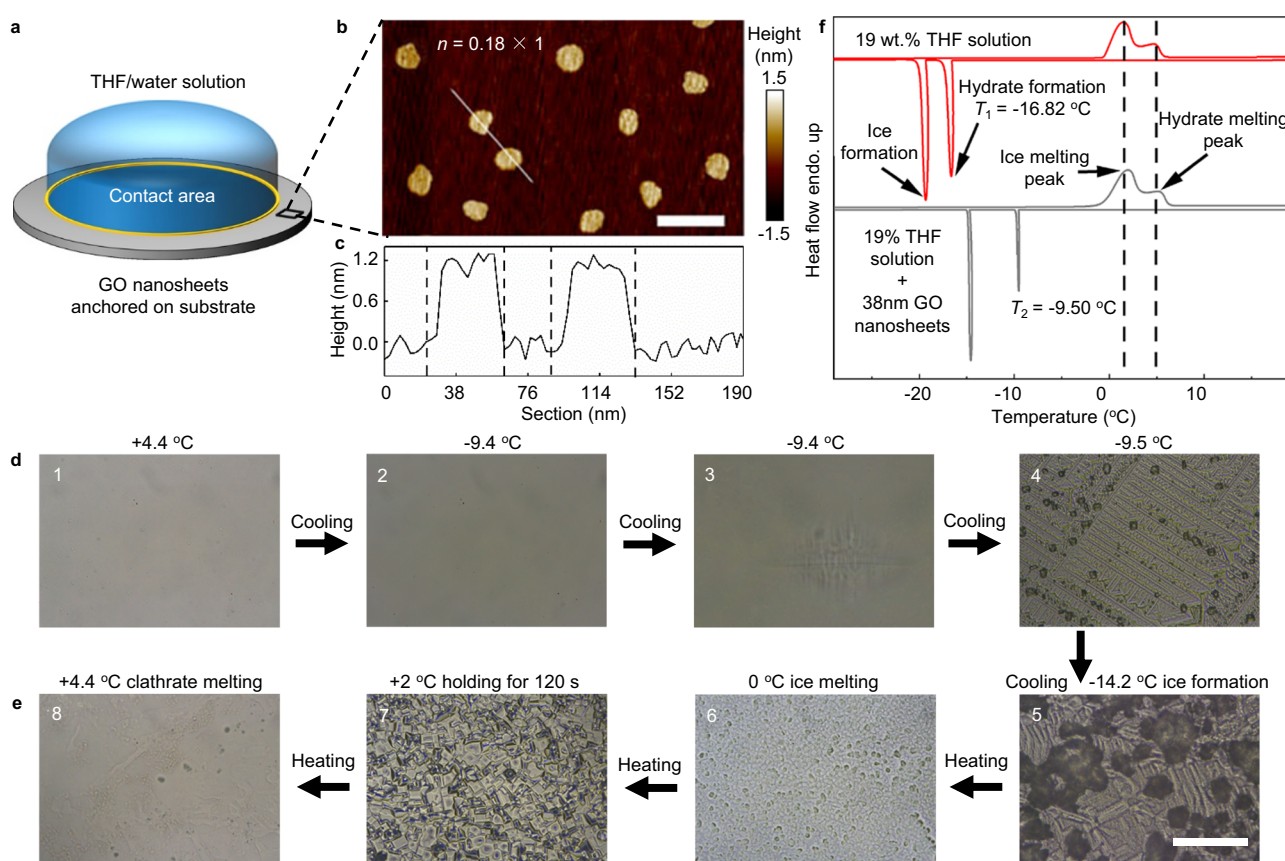

**Fig. 1 | The freezing behaviors of THF solution on the GOs anchored glass substrates. a** Schematic depiction of the THF/water solution deposited on the substrate anchored with GO nanosheets. The blue volume represents the drop of THF solution and the contact area between liquid drop and solid substrate is marked in the yellow circle. **b** shows the AFM images of 38 nm GOs (average lateral size) anchored on the substrate with the scaled coverage $n = 0.18 \times 1$ (relative GO graft density times scaled contact area). Scale bar, 100 nm. **c** shows the corresponding height profiles through the cross-section along the marked lines.

**d, e** Optical microscopy observation of the formation of THF clathrate (needle-like crystals, step 3) and ice (black and block-like substance, step 5) on the GO anchored surface under microscope in cooling (**d**) then heating (**e**) stages. Cooling rate, 1 °C min⁻¹. Sample volume, 3.5 μL; the scaled contact area of solution drop, $S = 1$ in the unit $S_0 = 0.25$ mm²; scale bar, 100 μm. **f** Thermal analysis of the formation and melting of THF clathrate hydrates and ice on the glass substrate surface without (red line) and with (gray line) the anchored 38 nm GO nanosheets by DSC.

Methods, Supplementary Figs. 1 and 2). The GO nanosheets with controlled sizes were chemically anchored on the APTES-modified glass surfaces as the substrate surfaces. This approach, anchored nanosheets on surfaces instead of directly dispersing them in drops, ensured a more accurate determination of the density and number of the nanosheets as the nucleation active sites while ensuring that nucleation occurred on the surface[29,42]. The GO density on surfaces was adjusted by changing the grafting time (Supplementary Tables 1–5); the contact area between the liquid drop and the substrate surface was determined by the diameter of liquid drop, which equals to the diameter of the cell (Fig. 1a). For example, Fig. 1b displays the morphologies of the substrate surface anchored with the 38 nm GO nanosheets with a specific graft density $D$ being 0.18 in the units of the graft number density $D_0 = 64\ \mu m^{-2}$. The contact area of liquid drop with substrate surface $S$ is recorded in the unit of $S_0 = 0.25\ mm^2$; thus, the number of GO nanosheets as the active nucleation sites, by accounting then estimated from AFM image (Fig. 1b), is $n$ in the unit of $n_0 = D_0 \times S_0 \approx 1.6 \times 10^7$ in this paper. The height of the grafted GO nanosheets on the substrate surface is uniformly about 1.2 nm (Fig. 1c), indicating the flat anchoring of GO nanosheets on surfaces (Supplementary Fig. 3).

$$n = D \times S \qquad (1)$$

The freezing behavior of a THF/water solution on the GO anchored surface in a changing temperature cycle (cooling from 30 °C to −40 °C with the constant cooling rate 1 °C min⁻¹ typically, then heating to 30 °C and keeping for 30 min before next cycle[32,43]) was monitored by optical microscopy (Fig. 1d, e) and a differential scanning calorimetry (DSC) (Fig. 1f, and Supplementary Fig. 2). The 30-min period at 30 °C is designed to eliminate the "memory effect" of the melted THF clathrate[32,43], as shown in Supplementary Fig. 2. When cooling the THF/water solution (on the surfaces anchored with GOs having the size of 38 nm and the GO coverage $n \approx 0.18$), as shown in Fig. 1d, the clathrate nucleation temperature on the surface was identified to be −9.4 °C as the crystal firstly appears during the cooling process (see Supplementary Discussion), which was determined by the consecutive optical microscopic observation[44,45] (Supplementary Movie 1). The remaining water (about 5.8% of the solution in weight, due to the evaporation of some THF molecules from the solution into the empty space of the sealed cell, see Supplementary Figs. 1, 2, and 4) occurred the ice nucleation, which was inferred by the sudden change of the opacity, at a lower temperature of −14.2 °C[46]. For more clearly checking the source of the remained water and thus the formed ice after the formation of the hydrates, the solutions with various ratios of THF to water (not rightly equal to that in the THF clathrate hydrates 1:17) were also applied in the experiments (Supplementary Fig. 4). More or less ice forms, corresponding to the smaller or larger THF-water ratio, verifying that the evaporation of THF molecules more than 1:17 water molecules from the solutions into the (small) empty space of the sealed cell is the reason; that is, there are some remained water to form ice while the concentration of the THF solution is rightly equal to (or only slightly larger than) that of hydrates. The small amount of THF and water evaporation into the empty space of the cell does not affect measuring the nucleation of the THF hydrates (more details see Supplementary Fig. 4).

The melting process was also observed with obvious morphological changes at 0 °C and 4.4 °C, corresponding to the melting of the ice and the THF clathrate hydrate respectively, where the found melting temperature of the clathrate, 4.4 °C, is in agreement with that in the ref. 47. As shown in Fig. 1f, two crystallization sharp peaks were observed in the DSC thermographic at −9.5 °C and −14.5 °C, as well as two melting broad peaks were observed at 0 °C and 4.4 °C, consistent with the direct optical microscopic observation showing the ice and THF clathrate formed separately[38] (More information about the DSC

analysis of clathrate and ice formation shown in Supplementary Fig. 5). As a comparison, the sharp crystallization DSC peaks of the THF solution on the substrate surface without anchored with any GO nanosheets were found at lower temperatures, about −16 °C (the clathrate formation) and −18 °C (ice formation), showing the facilitating ability of the 38 nm GO nanosheets on both the clathrate and the ice nucleation. The consistency between the DSC results and the optical microscopic observation verifies the effectiveness of our experiments in identifying clathrate nucleation and determining clathrate nucleation temperatures of THF/water solution drops from only the optical microscopy in the further experiments. In optical microscopy, the formation of THF clathrate as transparent, needle-like crystals[48] and the ice a black and block-like substance with a sudden change of opacity. When the two crystallization phenomena appeared, the corresponding THF hydrate and ice nucleation temperature were recorded.

## Size-dependent GO activity on the nucleation of THF clathrate

The effect of the GO size on the clathrate nucleation temperature was further investigated (Fig. 2a). We find that the THF clathrate hydrate shows an average nucleation temperature around −15.5 °C on the surfaces anchored with GOs of the lateral size being smaller than 31 nm. The nucleation is actually found to be almost independent on the presence of GO and its coverage (i.e., the relative number of GOs contacted with liquid drop $n$), this implies that the clathrate nucleation in this case is mainly triggered by the glass substrate considering that the homogeneous nucleation of THF clathrate was determined at a lower temperature around −32 °C[32,38]. Note that, when the size of anchored GOs increases from 31 nm to 38 nm, the average nucleation temperature displays an abrupt increase up to 6 °C from around −15.5 °C to about −9.5 °C (the GO coverage $n \approx 0.18$ and the cooling rate of 1 °C min⁻¹). Similar abrupt change appears using different GO coverages and different cooling rates (Supplementary Fig. 6), displaying the direct triggering clathrate nucleation by the anchored 38 nm GOs. While using larger GOs than 46 nm, the clathrate nucleation temperature does not further obviously rise (at the same GO coverage and cooling rate).

## Transitions in the nucleation activity of GO nanosheets

In a previous work[30], a similar transition of the nucleation temperature of ice was found to occur at a specific value of $L\ \Delta T_{ice}$ (the GO size multiplying supercooling) rather than a specific size ($L$) of GOs; this can be well explained as the transition occurs generally at a relative value ($l$ in Eq. 2) of the specific value size ($L$) of GOs to the diameter of critical ice nucleus $2R_c$, since $R_c$ is approximately inversely proportional to the supercooling temperature $\Delta T_{ice}$ based on the common applied classical nucleation theory.

$$l = \frac{L}{2R_c} \qquad (2)$$

For the clathrate nucleation events with various cooling rates from 1 to 8 °C min⁻¹ and different coverages of these GO nanosheets anchored on the glass substrates, we plot the averaged nucleation temperature ($\bar{T}$) as a function of $L\ \Delta T$ in Fig. 2b. Here $L$ is the average lateral size of GO nanosheets and $\Delta T$ is the corresponding supercooling temperature of the nucleation of the THF hydrate, with $T_m = 4.4\ °C$ being the equilibrium melting temperature of THF clathrate hydrate[49].

$$\Delta T = T_m - \bar{T} \qquad (3)$$

An abrupt change (transition) of the average nucleation temperature is found to always occur right at $L\ \Delta T \approx 650\ nm\ °C$ with the relative error about 10% (Fig. 2b), although the values of both $\Delta T$ and $L$

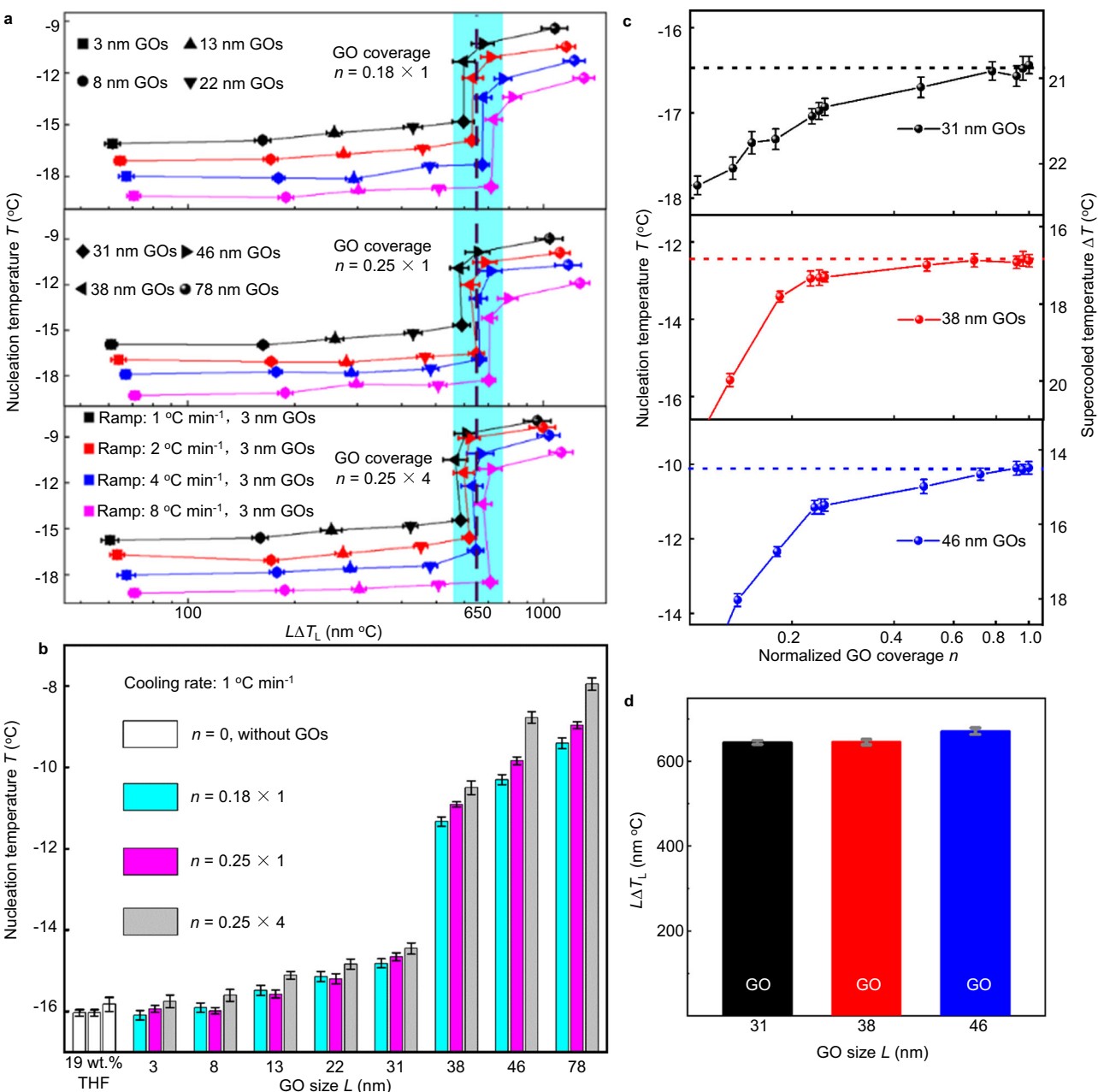

**Fig. 2 | Transitions in the THF clathrate hydrate nucleation activity of GO nanosheets at a specific value of the product of GOs' size and the supercooling.** **a** Relationship between the average nucleation temperature and $L\,\Delta T$ for the THF clathrate hydrate, and the nucleation temperature displays one abrupt change around $L\,\Delta T \approx 650$ nm °C for various GO nanosheets with different coverage, $n = 0.18$, $0.25$, and $1$, and under different cooling rates (Black-ramp of 1 °C min$^{-1}$, Red-ramp of 2 °C min$^{-1}$, Blue-ramp of 4 °C min$^{-1}$, Magenta-ramp of 8 °C min$^{-1}$). Black – cooling rate 1 °C min$^{-1}$, red – cooling rate 2 °C min$^{-1}$, blue –cooling rate 4 °C min$^{-1}$, and magenta – cooling rate 8 °C min$^{-1}$. Each nucleation temperature was averaged from 100 independent experiments. Error bars are the standard error of the mean (SEM). Data are shown as mean ± SEM. **b** The average nucleation temperature of THF clathrate tuned by GOs with various average sizes and different coverage of GO nanosheets. The white histograms are the glass substrates without anchored GOs. Cyan – GO coverage of $n = 0.18$, magenta – GO coverage of $n = 0.25$, and gray – GO

coverage of $n = 1$. Each nucleation temperature on the substrate without GOs was averaged from 85 independent experiments. Data are mean ± SEM. Each nucleation temperature on the substrate with GOs was averaged from 100 independent experiments. Data are mean ± SEM. **c** The THF nucleation temperature raises as increasing the coverage of GOs with the lateral size $L = 31$ nm, $38$ nm and $46$ nm, respectively, but has a size-dependent saturated value $T_L$ (the dashed lines). Black – 31 nm GO, red – 38 nm G, blue –46 nm GO. Each nucleation temperature was averaged from 100 independent experiments. Data are shown as mean ± SEM. **d** The product of the saturated supercooling nucleation temperature $\Delta T_L = T_m - T_L$ with the different GO sizes ($L$) of s is almost a constant, equal to 650 nm °C (the dashed line), for all the GOs with various $L$. Black – 31 nm GO, red – 38 nm G, blue –46 nm GO. Each nucleation temperature was averaged from 100 independent experiments. Data are mean ± SEM.

occurring the transition may change by varying the converge of GOs and the cooling rate, see Supplementary Fig. 6. When $L\,\Delta T < 650$ nm °C, the clathrate hydrate nucleation is almost not affected by the presence of GOs; when $L\,\Delta T = 650$ nm °C, the nucleation temperature

abruptly becomes higher, and then approximately remains unchanged at larger values when $L\,\Delta T > 650$ nm °C.

This indicates that the found transition on the facilitating capability of GOs on THF nucleation may occur for any size of GOs

(via varying the supercooling) rather than a specific size of GOs between 31 and 38 nm. We check this expectation by employing the same GO nanosheets but with varying graft density, of GOs and/or the contact area of solution with the substrates thus changing the coverage of GOs as nucleation active sites. As shown in Fig. 2c, we find that the average nucleation temperature first raises with the increased $n$ (coverage of GOs) but reaches a saturated value $T_L \approx -16.4, -12.6$, and $-10.2\,°C$ (the corresponding supercooling $\Delta T_L \approx 20.8, 17.0$, and $14.6\,°C$) for the GOs with the size $L = 31, 38$, and $46$ nm, respectively.

$$\Delta T_L = T_m - T_L \tag{4}$$

The remarkable result shows that GOs can sufficiently facilitate the hydrate nucleation while $\Delta T > \Delta T_L$ (i.e., $T < T_L$); whereas GO nanosheets do not have an obvious influence on the formation of clathrate hydrate as $\Delta T \leq \Delta T_L$ (i.e., $T > T_L$). As shown in Fig. 2d, we find that $L\,\Delta T \approx 650$ nm °C for these GOs with all the three sizes. It verifies that the capability of GOs in facilitating hydrate nucleation has generally a transition at the specific value of $L\,\Delta T$ rather than at a specific size of GOs or a specific supercooling.

We further measured the delay time (induction time, $t_D$) of hydrate nucleation as a function of both the supercooling and the (relative) number of the applied GOs (the top in Fig. 3a–c) to get the nucleation rate of the hydrate.

$$J = \frac{1}{n t_D} \tag{5}$$

Here $J$ is the nucleation rate of the THF hydrate induced by a unit number of GOs as the active site, which should be a function of only temperature and independent on the number of the applied GOs ($n$). Therefore, we expect that the scaled delay time $\tau$, which corresponds to the delay time of hydrate nucleation while applying a unit number of GOs as nucleation sites thus $\tau = J^{-1}$, should be a function of only $T$ rather than $n$ while the measured $t_D$ depends on both the $n$ and $T$.

$$\tau = n t_D \tag{6}$$

The relationship not only provides robust check for the measurement of the delay time, but also helps to get the nucleation rate of hydrate $J(T)$ in a wider temperature range by consisting of the multiple measurements of various n in experiments. As the 31 nm, 38 nm and 46 nm GO samples were each used with different values of $n$, while the delay time is indeed found to be dependent on both the $n$ and the temperature, as expected, $\tau$ is verified as a function of only the temperature, and independent of the GO coverage $n$ within experimental errors, see Fig. 3a–c. Again, we found that the function $\tau(T)$ demonstrates an abrupt change right at the $\Delta T_L$ found in the cooling experiments shown in Fig. 2c, (also see Supplementary Fig. 7, about 20.8, 17.0, and 14.6 °C) for the three kinds of GO nanosheets with various average lateral size (about 31, 38, and 46 nm), respectively. It shows well agreement in the expectation that the nucleation rate of hydrates on a unit number of nanosheets, $J(T) = \tau(T)^{-1}$ (Fig. 3d) has a derivative discontinuity at $\Delta T_L$ for all the three GO nanosheets with various sizes. Note that the transition of nucleation efficiency of GO nanosheets with three different sizes $L$ always occurs at their corresponding supercooling $\Delta T_L$ respectively, and satisfying $L\,\Delta T_L \approx 650$ nm °C with about 10% in deviation.

## Transitions in clathrate nucleation activity of nanocubes

We further used Au and Ag metal nanocubes of controlled sizes to verify the relationship between the nucleation activity and the formation of THF clathrate (Fig. 4, Supplementary Figs. 8 and 9). The Ag nanocubes with side length $L = 45$ nm (Fig. 4a–c) and Au nanocubes

with $L = 45$ nm (Fig. 4d–f), 70 nm (Fig. 4g–i) were anchored on glass substrates, respectively; and the graft density of the particles were controlled. With varying the graft density of nanocubes as nucleation active sites ($n$), particle size, and type, we found a similar result to that with GOs, that is, $\tau = n t_D$ is independent of the Au/Ag nanocube coverage within experimental errors (Fig. 4j, and Supplementary Fig. 9); and the $\tau(T)$ has an abrupt change at $\Delta T_L \approx \frac{450\,nm}{L}\,°C$ in all cases. Thereby the clathrate nucleation activity of all these Au and Ag nanocubes exhibits a transition at $L\,\Delta T \approx 450$ nm °C (inset in Fig. 4j).

The generally found abrupt change of the THF hydrates nucleation on nanoparticles at a specific value of $L\,\Delta T$ corresponds well to the abrupt value change of the nucleation rate

$$J = A e^{-\Delta G^*/(k_B T)} \tag{7}$$

The kinetic factor ($A$) is less affected by the temperature and the nanoparticle size, a fortiori, it is less possibility to have a transition at a specific value of $L\,\Delta T$. Here $k_B$ is the Boltzmann constant. Noted that the other factors, such as the different heat conductance for nanoparticles of various sizes, and the dynamics of molecules neighboring nanoparticles (see Methods), are not possible to contribute to the observed transition occurring at the specific value of $L\,\Delta T$ under all these different experimental conditions, especially the delay time measurements with fixing both the size of nanoparticles and supercooling. Thus, we concluded that the transition of nucleation rate at a specific $L\,\Delta T$ is ascribed to that of the nucleation free-energy barrier $\Delta G^*(\Delta T; L)$. This suggested a general source about the discontinuity of the nucleation free-energy barrier (in forming critical hydrates nucleus on finite-sized nanoparticles). Similar result was also found in the ice nucleation on controlled-size nanosheets[29] where the nucleation rate had a transition at a specific value of the product of water supercooling and size of nanosheets. These general findings in different nucleation processes on various substrates can be well understood based on a simple assumption that the interface between the (critical) nucleus and its surroundings (liquid water/solution) has (approximately) an optimized shape to minimize its interface free energy. The assumption was reasonable especially for the cases where the interface tension is significant thus the dominant pathway of nucleation is to optimize the shape of nucleus and the nucleation free energy. Note that such an assumption was one in the classical nucleation theory (CNT). Under the assumption without requiring the others in CNT, we can very generally give that the curvature radius of the interface between the critical nucleus and its surroundings is shown as

$$R_c = \frac{2\gamma}{|\Delta\mu|} \tag{8}$$

Here the interfacial energy between the critical nucleus and its surrounding liquid environment, $\gamma$, is less dependent on the supercooling $\Delta T$, thus it is approximated as a constant within a not large range of $\Delta T$ in practice; while the volume free energy difference of the (microscopic) critical nucleus and the surroundings is approximately proportional to the supercooling since it disappears at $\Delta T = 0$. At this condition, we have

$$\Delta\mu \approx \Delta S \Delta T \tag{9}$$

Here the $\Delta\mu$ is chemical potential, and the $\Delta S$ is the entropy. With an approximate constant $\Delta S$ for not large $\Delta T$, thus we have

$$R_c \approx \frac{2\gamma}{|\Delta S|}\frac{1}{\Delta T} \tag{10}$$

with the approximated constant coefficient $\frac{\gamma}{|\Delta S|}$. It is worth mentioning that the inverse-proportional relationship between the radius of

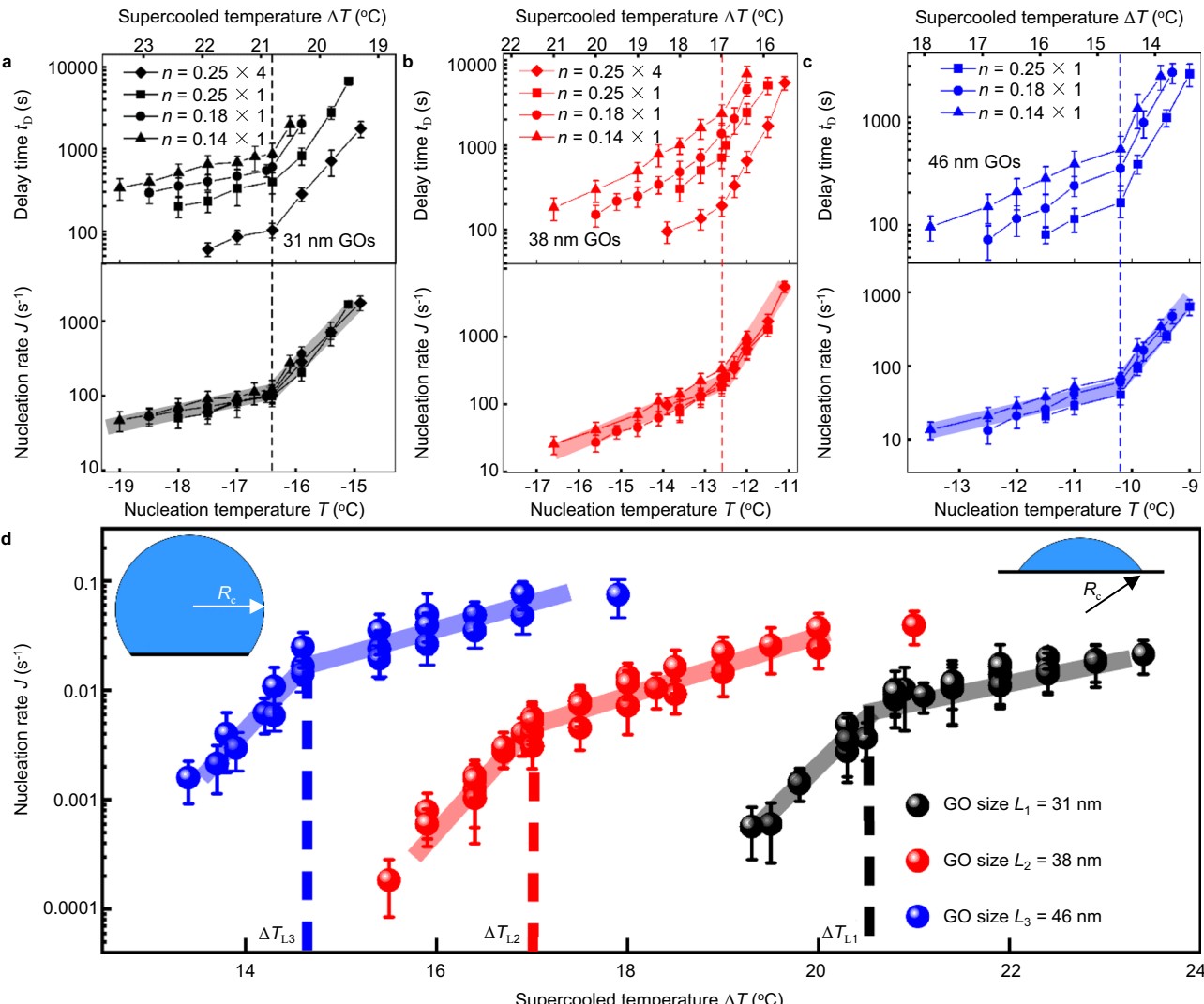

**Fig. 3 | Transitions in the nucleation rate during the THF clathrate formation on 31 nm, 38 nm and 46 nm GO nanosheets anchored glass substrates, respectively. a–c** The top panels show the (average) delay time $t_D$ of clathrate hydrates of THF-water mixture as the applied constant temperature (and corresponding supercooling shown in top $x$-axis) for a few $n$ values (the relative number of GO nanosheets as the nucleation sites), and the bottom panels show $\tau = nt_D$ for the same data in the top, as the GO size of 31 nm (black), 38 nm (red), 46 nm (blue), respectively. Abrupt changes (transitions) of the THF clathrate hydrate nucleation activity of GO nanosheets at specific supercooling value depending on the size of applied GO nanosheets. Every average clathrate hydrate nucleation delay time in (**a–c**) shows mean ± SEM. For the GO coverage of $n = 0.14$, the mean values were averaged from 36 measurements. For every other GO coverage ($n = 1$, $n = 0.25$ and $n = 0.18$), the mean values were averaged from 39 measurements. **d** The $J = \frac{1}{\tau}$ gives the nucleation rate of the clathrate on the unit number of GO nanosheets. Black – 31 nm GO, red – 38 nm G, blue – 46 nm GO. The insets show the illustrations of the critical clathrate nucleus on nanosheets as the size of GO nanosheet is smaller (top-left) or larger (top-right) than the spherical diameter of critical clathrate nucleus $2R_c$, respectively, based on the classical nucleation theory. Every average clathrate hydrate nucleation delay time in (**d**) is processed from (**a–c**), and shows mean ± SEM. For the GO coverage of $n = 0.14$, the mean values were averaged from 36 measurements. For every other GO coverage ($n = 1$, $n = 0.25$ and $n = 0.18$), the mean values were averaged from 39 measurements.

critical nucleus and the supercooling is more general than the CNT although it was usually derived via the CNT in literature[18,19,33].

The dependence of the free-energy barrier on the size of nanoparticles $L$ should be written as a function of a dimensionless variable about the size of nanoparticles, as shown in Eq. (2), since $2R_c$ is the characteristic length here corresponding to the nucleation free energy barrier. We write the equation of

$$\Delta G^*(\Delta T, L) = \Delta G^*(\Delta T, \infty)g(l) \qquad (11)$$

Here $\Delta G^*(\Delta T; \infty)$ is the normal heterogeneous nucleation free-energy barrier on sufficiently large plane of substrates, and the function $g(l)$, which corresponds to the correction of the finite size of nanoparticle surfaces on facilitating the nucleation, is found to indeed

have a derivative discontinuity at $l \approx l_c$ a specific value of the dimensionless size of nanoparticles, via the CNT. As sketched in the inset of Fig. 4k, while $l > l_c$ (large substrates), it is the same as the normal heterogeneous nucleation (on infinite-size flat surfaces) where the critical nucleus is a spherical cap with a small contact angle (determined by the interaction of substrates with the nucleating molecules) sitting on the substrates. By contrast, when $l < l_c$, the nucleus changes its shape after meeting the edge of nano-size substrates (nanoparticles) to be a spherical cap (on circular nanosheets, see inset of Fig. 3d) or a square-bottom spherical cap (on the nanocubes, see inset of Fig. 4k) with a large contact angle by pinning at the edge of substrates but keeping its interface with surrounding solution close a spherical surface to optimize its surface energy. It leads to a larger free-energy barrier of the nucleation on smaller substrates than that on

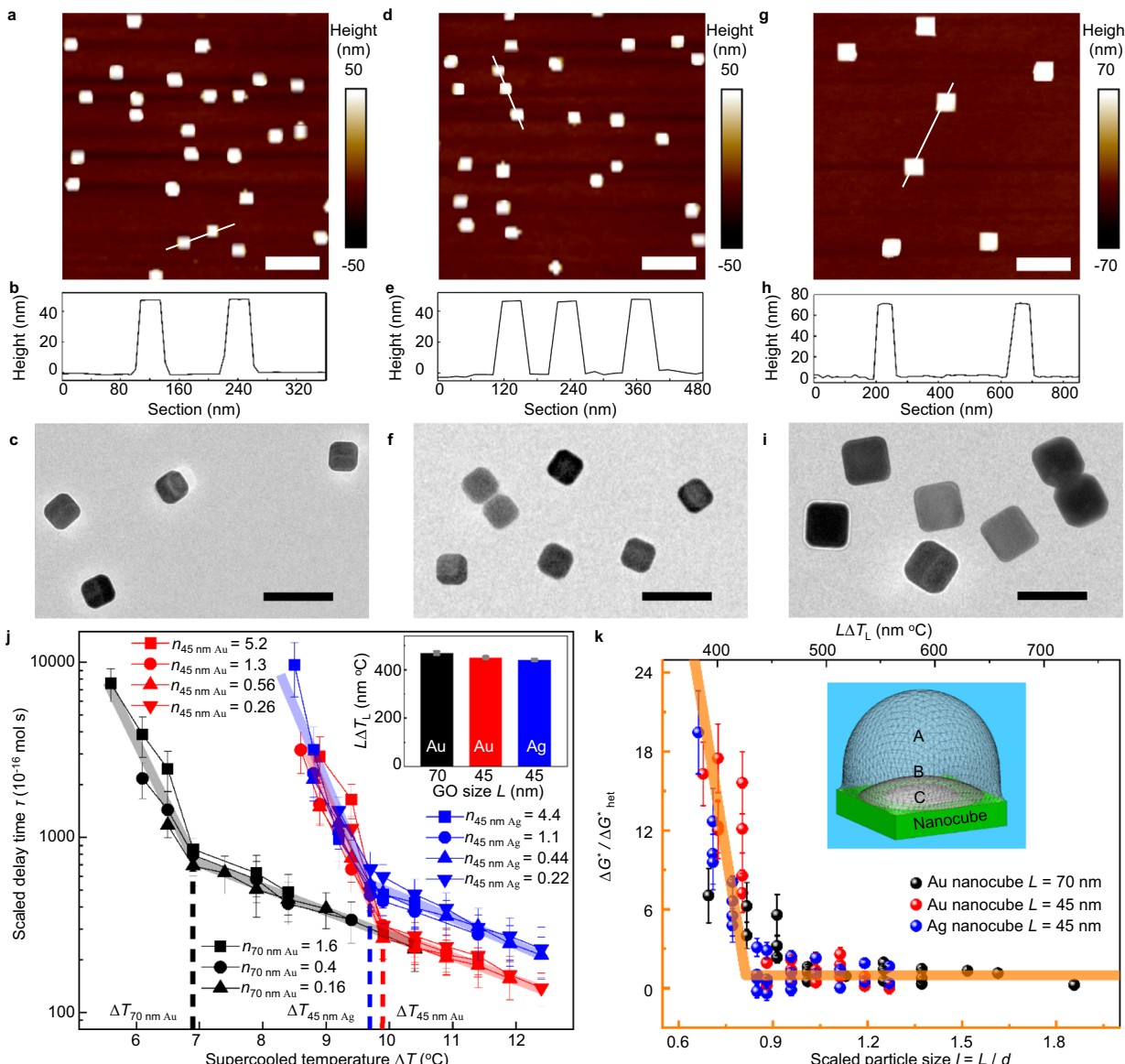

**Fig. 4 | Abrupt changes (transitions) of the THF hydrate nucleation activity of Ag/Au nanocubes at specific supercooling temperature $\Delta T_L$ which is determined by the size of nanoparticles $L$.** **a**–**i** The AFM images, corresponding height profiles through the cross-section along the marked lines of anchored nanocubes on the substrate, and TEM images of suspended nanocubes. 45 nm Ag nanocubes (**a**–**c**), 45 nm Au nanocubes (**d**–**f**), and 70 nm Au nanocubes (**g**–**i**), respectively. Scale bar of (**a**), (**d**) and (**g**) AFM images, 200 nm. Scale bar of (**c**), (**f**) and (**i**) TEM images, 100 nm. **j** The scaled delay time of nucleation of THF aqueous solution, $\tau = nt_D$, versus $\Delta T$. The three curves for each nanocube come from different $n$ and collapse into the same curve. Black – 70 nm Au nanocube, red – 45 nm Au nanocube, blue – 45 nm Ag nanocube. The inset shows the $L \Delta T_L \approx 450$ nm °C for all the Au and Ag nanocubes. Magenta – 70 nm Au nanocube, cyan – 45 nm Au nanocube, orange – 45 nm Ag nanocube. Every average clathrate hydrate nucleation delay time shows mean ± SEM. The mean values were averaged from 36 measurements for the 70 nm Au nanocube with coverage of $n = 0.16$, 45 nm Au nanocube with coverage of $n = 0.26$, and Ag nanocube with coverage of $n = 0.22$. For the every other nanocube coverage, the mean values were averaged from 39 measurements. **k** The free-energy barrier of THF hydrate nucleation $\Delta G^*$ on the Ag/Au nanocubes in the unit of that on sufficient-large substrates $\Delta G^*_{het}$. The free-energy barrier obtained from the data in Fig. 4j is compared with the theoretical calculation based on the spirit of CNT (orange line and axis, the transition occurs at $l_c = \frac{L}{2R_C} \approx 0.8$ for the nanocubes with the side length $L$, see Supplementary Fig. 8). Black – 70 nm Au nanocube, red – 45 nm Au nanocube, blue – 45 nm Ag nanocube, orange – fit line and corresponding XY axis. Inset, schematic illustrations of the shapes of clathrate nucleus with various sizes on a nanocube. Every data point in (**k**) is processed from (**j**), and shows mean ± SEM. The mean values were averaged from 36 measurements for the 70 nm Au nanocube with coverage of $n = 0.16$, 45 nm Au nanocube with coverage of $n = 0.26$, and Ag nanocube with coverage of $n = 0.22$. For the every other nanocube coverage, the mean values were averaged from 39 measurements.

larger substrates. It is worth mentioning that the pinning of nucleus at edge of substrates does not require any specific interaction between the nucleus and the edge of substrates, but the result of requiring to optimize the shape of nucleus then the surface free energy, thus it is general for various substrates and nucleating molecules. Therefore, as shown in Fig. 4k, there is a derivative discontinuity of the free-energy barrier of nucleation on finite-sized substrates at $l = l_c$, and $l_c$ is

dominated by geometry of the finite-size substrates and less depends on other aspects of substrates such as the interaction between the substrates and (THF/water) molecules. For square-shape substrates (nanocubes), we theoretically have $l_c \approx 0.8$ (Supplementary Fig. 8); but for circle-shape substrates (GO nanosheets), $l_c \approx 1.0$. Therefore, for the Au/Ag nanocubes, the found transition condition, $L \Delta T \approx 450$ nm °C which is smaller than that in GOs nanosheets, is due to the different

shape of nanocubes from the nanosheets to bring the different value of $l_c$. We estimate that the spherical radius of the critical nucleus of the THF hydrates $R_c \approx \frac{450\,°C}{2l_c \Delta T}$ nm $\approx \frac{280\,°C}{\Delta T}$ nm. The value is still slightly smaller than the result on GO nanosheets, $R_c \approx \frac{325\,°C}{\Delta T}$ nm, if supposing the GO nanosheets being circles then $l_c \approx 1.0$. The difference may partially come from the deviation of the GO nanosheets to the perfect circles. Considering all of those, we have the radius of critical nucleus of the THF clathrate hydrates $R_c \approx \frac{300\,°C}{\Delta T}$ nm with an error about 10–20%, indicating that the constant of the THF hydrates $\frac{\gamma}{|\Delta S|} \approx 150$ nm °C. Especially, the size of critical nucleus, $R_c \approx \frac{300\,°C}{\Delta T}$ nm inversely proportional to the supercooling temperature $\Delta T$, is in well agreement with the expectation of CNT. Therefore, we actually present a nanometer ruler (nanosheets or nanocubes with a specific size) and detect the nucleation signal on it. When the size of critical nucleus (dependent on $\Delta T$) matches with that of the nanometer ruler, the clathrate forms thus the size of critical nucleus can be recorded. It is worth mentioning that current work about THF clathrate nucleation gives the size of critical nucleus of THF hydrates being about a few larger than that of the ice nucleus[29], which is calculated and reported by quantitative analysis.

## Discussion

As we already mentioned, the general experimental finding is well explained via assuming that the clathrate hydrate is controlled by the formation of the critical nucleus with optimized interface shape on nanoparticles. We can achieve the value of $\frac{\gamma}{|\Delta S|}$ of the clathrate hydrate nucleation for comparing with both macroscopic measurements and molecular simulations, providing some insights about the microscopic character of the critical nucleus and its surrounding solution as well as the similarity and/or difference from their macroscopic counterparts (*i.e.*, the clathrate hydrate crystals and solutions, respectively). So far, there is no very solid data about both the melting entropy of macroscopic clathrate hydrate crystals and the surface tension between the crystals and solutions. For the THF sII hydrates, the macroscopic melting entropy was roughly estimated to more than 70 kJ$^{-1}$ °C$^{-1}$ mol$^{-1}$ [38,47,50]. Considering that the density of THF hydrate crystal to be $\rho \approx 0.99$ g cm$^{-3}$ [47] and the molar mass of THF hydrate involving 17 water molecules is 378 g mol$^{-1}$, we show a $|\Delta S|$ of more than 780 mJ$^{-1}$ °C$^{-1}$ cm$^{-3}$. The measurement of surface tension of clathrate hydrates is more difficult. A reported estimate about the $\gamma$ of the sII clathrate hydrate of propane about 25 mJ m$^{-2}$ could provide a reference about that of the THF hydrates[51]. Further considering the estimates and measurements of various hydrates in references[19,51–53], we might assume that the surface tension of the THF hydrates with its aqueous solution $\gamma$ is $\approx 30$ mJ m$^{-2}$, in the same order of that between ice and water. Thus, we roughly estimated the macroscopic value of the THF hydrates $\frac{\gamma}{|\Delta S|} \lesssim 40$ nm °C. This estimate is consistent with the recent results about the methane clathrate by MD simulations[19], where $R_c \Delta T \approx 73.2$ nm °C, i.e., $\frac{\gamma}{|\Delta S|} \approx 36.6$ nm °C, by choosing 36 mJ m$^{-2}$ as the surface tensions between liquid and sI methane clathrate crystal, but much smaller than the obtained microscopic value obtained from our current experiment, about 150 nm °C. In other side, a recent publication reported the cryo-SEM observation of the formation of 10–30 nm nano-clusters of THF clathrate hydrate at a supercooling level of approximately 20 °C, before the appearance of THF clathrate crystals[33], which is consistent with our results, as the critical nucleus diameter under 20 °C supercooling is approximately about $\Delta L = \frac{600\,\text{nm}\,°C}{20\,°C} = 30$ nm. Therefore, besides the possible difference in the estimated values of the melting entropy and surface tension of the THF clathrates from that of gas clathrates, as well as the possible deviation between simulations and experiments, the large critical nucleus of THF clathrates in the current experiments may indicate the difference of the nanoscale THF hydrate nuclei/surrounding solution from their macroscopic counterparts. Further investigation into the static structure and dynamic behavior of hydrate critical nuclei will yield additional experimental evidence for understanding the interactions

between the nuclei and surrounding solutions at the nanoscale[22,54–56]. The conclusion, which still needs further verification by more simulations and experiments, is consistent with the findings in MD simulations about the occurrence of some pre-nucleation steps which might lead to different structures and character of surrounding solution of critical nucleus from the macroscopic solution[22].

An intriguing and still open question is the comparison between the nucleation mechanism of the THF hydrates and the gas hydrates. A key distinction likely arises from the different mass transfer dynamics of guest molecules, involving the transfers of the dissolved guest molecules from the surrounding solution to form the clathrate hydrate nuclei, and the transfer of guest molecule from the gas phase into the solution for the apparent clathrate growth (specifically for gas clathrates). For the formation of nanoscale-size critical nuclei, it is mainly related to the local transferring kinetics of the dissolved guest molecule. In the case of gas clathrate, the low concentration of dissolved gas molecules in the solution results in a number ratio between gas guest and water molecules smaller than that in the formed clathrate, which may induce different effects from the process of forming THF hydrates from a miscible THF solution. It is worthwhile to investigate the possible effects in the formation of critical nuclei in the future.

In conclusion, our experimental study successfully probes the critical nucleus size in THF clathrate hydrates and reveals a crucial relationship between its size and the supercooling level. By demonstrating the inverse proportionality of the critical nucleus size to the degree of supercooling, we contribute valuable insights into the microscopic mechanism of clathrate hydrate nucleation. Unlike typical research focused on detecting early stages of the formation of clathrate hydrate through molecular simulations[10–20], such as the formation of water nanocages encapsulating guest molecules, our work delves into the specific critical nucleus formation. The reported technique of using nanoparticles as the probe is demonstrated to have general feasibility in probing the occurrence and character of the (transient and small) critical nucleus in numerous of the 1st-order phase transition processes.

## Methods
### Preparation of materials
Tetrahydrofuran (THF, inhibitor-free, suitable for HPLC, purity $\geq$ 99.9%), (3-aminopropyl)-triethoxysilane (APTES, 99%), trimethoxysilylpropanethiol (MPTMS, 99%), sulfuric acid ($H_2SO_4$, 98%), hydrogen peroxide solution ($H_2O_2$, 30% in water), $N$-hydroxysuccinimide (NHS, 98%), and $N$-(3-ddimethylaminopropyl)-$N'$-ethylcarbodiimide (EDC, $\geq$ 97%) were purchased from Sigma-Aldrich (Merck). Ethanol (EtOH, $\geq$ 99.5%) was purchased from Beijing Hua Gong Chang (Beijing, China). Ultrapure water (18.2 M$\Omega$ cm) was produced by Millipore Milli-Q reference with a 0.22-μm membrane (Millipak-40). Au nanocubes (1 mg mL$^{-1}$) and Ag nanocubes (1 mg mL$^{-1}$) were procured from Beijing Zhongkeleiming Technology Co. Ltd. (Beijing, China). The aqueous dispersion of GOs (10 wt.%) with a broad size distribution was procured from LEADNANO Co., Ltd. (Jining, China). For the preparation of GOs of controlled sizes, we fractionated the GO nanosheets through filtration by employing various membranes[29]. The obtained GO fraction was dispersed in water and stored at 4 °C. The mass concentrations of GO aqueous dispersions were measured by drying the sample with a fixed volume and weighing the residue solid content of GOs.

### Characterization
Raw GO and metal nanocube materials were images with a transmission electron microscopy (TEM, JEM-2100F, JEOL) as shown in Supplementary Fig. 10 and Fig. 4. The structural defects of raw GOs were measured by Raman spectroscopy on a LabRAM ARAMIS spectrometer equipped with a 532 nm laser (HORIBA Jobin Yvon, France)[57,58] as shown in Supplementary Fig. 11. The morphology and thickness of anchored GOs and nanocubes on glass surfaces were investigated with

a Multimode 8 AFM (Bruker, Germany). The distributions of hydrodynamic diameters of prepared various GO samples were measured by dynamic light scattering (DLS, ALV-5022F) as shown in Supplementary Fig. 12. The elemental content of raw GO samples and modified surfaces were determined with X-ray photoelectron spectroscopy (ESCALab250-XL, VG, UK)[59,60] as show in Supplementary Fig. 13. A DSC 8500 differential scanning calorimeter (DSC) from PERKINELMER was used to probe the thermograph of hydrate formation and dissociation. Optical imaging of nucleation measurements was done with an Olympus X51 microscope, coupled with a high-speed camera (Phantom V7.3, VEO440L-18G, Vision Research, USA).

## DSC analysis

A THF/water solution at a composition of 19 wt.% (1:17 molar ratio of THF to water) without or with dispersed GO nanosheets of 4 mg mL$^{-1}$ was prepared. Next, in every test, the mixture samples of 3.0 μL were added into the aluminum crucible and rushed to being sealed. Then, the formation of THF clathrate hydrates was determined by using the thermal cycle (Supplementary Fig. 3): (step 1) holding the sample at 30 °C for 1 min; (step 2) cooling the sample to a low temperature of usually −40 °C with the cooling rate of 1 °C min$^{-1}$; (step 3) holding the sample at −40 °C for 1 min; (step 4) heating the sample to the temperature of 30 °C with the heating rate 5 °C min$^{-1}$. In the cooling, the crystallization peaks of THF clathrate hydrate and ice formation can be detected. Moreover, in the heating, the corresponding melting peaks of ice and clathrate hydrate can be shown. Before starting the DSC test, the calibration should be carried out, in which the temperature and heat were calibrated by checking the melting point and fusion enthalpy of indium with temperature accuracy of 0.1 °C[38]. Additionally, the sealing is verified using differential scanning calorimetry (DSC) with two loops of the cooling and heating process, during which the THF clathrate hydrates are formed twice, and both formations yield identical results (shown in Supplementary Fig. 2). Moreover, to ensure the integrity of the cell and prevent any leaks, we performed additional weight measurements before and after each nucleation experiment. The sample was weighed twice, and the difference between the two measurements was consistently within 0.02 mg. This meticulous procedure guarantees that the accuracy of our experimental results remains unaffected by any potential losses of THF and water during the course of the experiments.

## GOs nanosheets and metal nanocubes anchored on glass surfaces

Glass substrates (Linkam 3930) were cleaned and hydroxylated in the mixture solution, consisting of 98% $H_2SO_4$ and 30% $H_2O_2$ in volume ratio of 7:3 (v/v) at 90 °C for 30 min. After the thorough ultrasonic treatment with ultrapure water for three times, and the glass slices were immersed in a freshly prepared APTES solution (1 wt.% in EtOH) for different time periods. Next, the slices were taken out and sonicated in ultrapure water, generating the desired APTES-SAM[61]. Subsequently, the APTES-SAM-covered glass substrate was kept in the prepared GO aqueous dispersions with EDC-NHS with different time periods for obtaining APTES-GO film on glass surfaces of various coverage, and then the GOs anchored glass slices were ultrasonically cleaned in ultrapure water and blown dry with $N_2$. The obtained samples were labeled as glass-APTES-GOs with different GO sizes and various GO coverages.

## Au/Ag nanocubes anchored on glass surfaces

Glass substrates (Linkam 3930) were cleaned and hydroxylated in the mixture solution, consisting of 98% $H_2SO_4$ and 30% $H_2O_2$ in volume ratio of 7:3 (v/v) at 90 °C for 30 min. After the thorough ultrasonic treatment with ultrapure water for three times, and the glass slices were immersed in a freshly prepared MPTMS solution (5 wt.% in EtOH) for different time periods. Next, the slices were taken out and sonicated in ultrapure water, generating the desired MPTMS-SAM[62]. Subsequently, the MPTMS-SAM-covered glass substrate was kept in the prepared Au/Ag nanocubes aqueous dispersions with different time periods for obtaining MPTMS-Au/Ag nanocubes film on glass surfaces of various coverage, and then the Au/Ag nanocubes anchored glass slices were ultrasonically cleaned in ultrapure water and blown dry with $N_2$. The obtained samples were labeled as glass-MPTMS-Au/Ag nanocubes with different sizes and various coverages.

## Measurement of THF clathrate hydrate nucleation

The THF clathrate hydrate nucleation temperature ($T$) and delay time ($t_D$) were measured in a sealed sample cell consisting of a glass O-ring (height 1.5 mm, inner diameter changed from 2.5 to 5 mm) sandwiched between one glass substrate with anchored GO nanosheets (metal nanocubes) and one optical microscope cover glasses by the UV fast cure glue. Inside the closed cell, 3.5 μL THF/water mixture was dropped atop the modified glass substrate anchored with GO nanosheets (metal nanocubes) of different lateral sizes. The entire preparation of the sample cell was carried out in a Class II Type A2 biosafety cabinet to avoid contamination. Then the closed cell was placed atop a cryo-stage (Linkam LTS420) and the cryo-stage was calibrated as outlined in Ref. 63. The formation of THF clathrate hydrate and ice was observed through an optical microscopy. On the one hand, the temperature of the crystal morphology[44,45] first appearing, was identified as the hydrate nucleation temperature (shown in Supplementary Movie 1); on the other hand, the temperature at which a sudden change in the opacity[46] of samples after the formation of hydrate was regarded as ice nucleation temperature. In this method, the number of nucleation sites was tuned by the GO (metal nanocube) graft density and the contact area of the THF/water sample with the substrate. Moreover, for the used nanoparticles (including melt nanocube with size large than 45 nm, and GO nanosheets with size larger than 30 nm) in this study, the calculated temperature difference between the top and bottom surfaces of the nanoparticles is within 0.03 °C. This marginal temperature difference has no impact on the statistical analysis of nucleation temperatures, consequently, does not influence the conclusions drawn from the study. The data of THF hydrate nucleation temperature are the mean with the standard error of the mean (SEM). For each mean nucleation temperature, the total number of measurements is more than 80. We follow the mentioned method to estimate the mean delay time of THF hydrate nucleation[29]. In brief, the delay time at a certain temperature was measured as the time period elapsed from the time when the samples were cooled to a target temperature to the time when the THF clathrate hydrate nucleation occurred. We independently measured the clathrate hydrate nucleation delay time with more than 35 valid nucleation events.

## Reporting summary

Further information on research design is available in the Nature Portfolio Reporting Summary linked to this article.

## Data availability

The data that support the findings of this study are available from the corresponding authors upon request.

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

## Acknowledgements

H.X., L.Li., Y.L., K.C., Z.H., G.B., J.L., and J.W. were supported by the National Natural Science Foundation of China (Grant No. 21733010, 51925307, 22105210, 32001083, 52273220), National Key R&D Program of China (2018YFA0208502), Key Research Program of Frontier Sciences, Chinese Academy of Sciences (Grant No. ZDBS-LY-SLH031), Beijing National Laboratory for Molecular Sciences (Grant No. BNLMS-CXXM-2020BMS20 and the Strategic Priority Research Program of the Chinese Academy of Sciences (Grant No. XDB28000000). Y.W. and X.Z. were supported by the National Natural Science Foundation of China (Grant No.12174388 and T2293761).

## Author contributions

H.X., L.L., and Y.L. performed the nucleation experiments. H.X., Z.H., and K.C. performed the DSC experiments. H.X., Z.H., Y.L., L.L., G.B., Y.W., X.Z., J.L., and J.W. analyzed the data. H.X., J.L., X.Z., and J.W. prepared the manuscript. H.X., X.Z., and J.W. conceived the project and designed the experiments.

## Competing interests
The authors declare no competing interests.
