## [Peer Review File · Nature Communications]

Probing the critical nucleus size in tetrahydrofuran clathrate hydrate formation using surface-anchored nanoparticlesReviewers' comments:

Reviewer #1 (Remarks to the Author):

This study reports on the nucleation of THF clathrate hydrates on nanosheets and nanocubes. This study actually follows very closely to a prior study published by some of the authors in Nature in 2019 for ice. It is clear that the authors have done extensive work and characterization of the systems considered, which is quite important. There are several issues with the present manuscript, which diminish the impact and novelty of the study compared to the one published in Nature on ice.

* One of the main concerns here is the choice of clathrate hydrate system. It is understandable that THF was chosen as THF is miscible in water and the formation of clathrate hydrates is a bulk phase transition just as in ice. However, most clathrate hydrates of interest are formed from gas, which are not miscible in water and mass transfer becomes quite significant. For this reason, it is perhaps not surprising that the results reported for THF clathrate hydrates are qualitatively similar to those reported for ice in the Nature paper. As such, the authors should be very explicit that the findings from this study is very specific to THF clathrate hydrates and not for any other type of clathrate hydrates.

* One of the most difficult parts to understand is the results presented in Fig. 2, especially the use of DT.L as an independent variable to quantify the nucleation. This same variable is used in the Nature paper for ice. What is the meaning of this variable and what is so unique about it? What is no unique in going from 31 nm to 38 nm in the size of the GO to cause such change in the nucleation temperature.

* The other variable that is not well defined and explained is tau ($= n \cdot \text{delay_time}$). From tau, the nucleation rate is obtained. In the text, it is stated that J only depends on T, and not on n. How is this explained? The statements are inconsistent based on the data and equation shown.

* Another major concern in the results is that the DSC scans show the formation of ice. At the THF concentration used, which is the stoichiometric concentration for forming structure II clathrate hydrates, no water should be available to form ice. THF is of course volatile and should evaporate during the experiment. However, how can one be confident that the phase transition measured is actually that for THF clathrate hydrates and not ice?

* The data in Fig. 3 for the 38 nm GOs does not shown the clear breaks for the delay time and tau, which in panel b, surprisingly shows a clear separation of the data above and below the subcooling of 17 K.

Minor points:

* Units for the variables should be consistent. For example, some temperatures are in degree C and others in K.

* The details of what is DG* are said to be in the Methods section, but they are no where to be found.

* Some parts of the text needs improvement.

* The title is misleading and should be more specific to the system and conditions studied.

Reviewer #2 (Remarks to the Author):

In this manuscript, the authors report the nucleation and crystallization of THF clathrate hydrate. GO nanosheets with certain sizes can reduce the degree of supercooling in the THF clathrate hydrate formation. The reported data are quite interesting and they would give us new knowledge in the nucleation of THF clathrate hydrate. The methodology is also interesting. However, the reviewer feels the present manuscript needs major revisions before publication.

1. The most significant point the authors need revising is that the authors have never observed the existence of critical nucleus directly. The authors changed the size of GO nanosheets and measured

the induction time and the degree of supercooling in the THF clathrate hydrate formation. The authors should clarify why the authors can discuss the size of the critical nucleus on the basis of GO nanosheet experiments. Generally, the size of the critical nucleus should be smaller than 10 nm. The reviewer thinks the experimental results do not always support the conclusions.

2. Recently, Machida et al (CrystEngComm, vol. 24, pp. 6730 (2022)) reported the different nucleation mechanism of THF clathrate hydrate. They stated the mechanism is not the classical nucleation theory, but similar to labile cluster or blob mechanism. Though the results are different from the ones the authors suggested, there seems to be similar points on the sizes and/or the interaction between THF solution and materials (GO sheets or crystal seeds). At least, the author should discuss their results in the manuscript.

3. The results clearly indicate the product of the degree of supercooling and the size of GOs would be an index as THF clathrate hydrate nucleation. However, the reviewer hardly understand its scientific meanings. Moreover, when Ag (or Au) nanocubes were used instead of GOs, the index was also changed. The reviewer wants the authors to discuss the scientific meanings.

4. The authors should cite other recent articles on THF clathrate hydrate nucleation. For example, Huang et al., J.Phys.Chem.C, vol. 124, pp. 13966 (2020) and Cryst. Growth Des., vol. 20, pp. 2921 (2020), and Ghosh et al., J.Phys.Chem.C, vol. 123, pp. 16300 (2019).

Reviewer #3 (Remarks to the Author):

This manuscript investigates the formation of THF-water clathrate hydrate in the presence of nanoparticle coated surfaces. THF-water solutions are very topical as a model clathrate-forming system, as THF and water mix at the required composition and the crystallisation temperature is around 277K at ambient pressure. This contrasts for example with methane-hydrates, where nucleation and growth are limited by the gas solubility and surface area of the gas-liquid interface.

Here, surface-anchored graphene oxide (GO) and Au/Ag nanoparticles with controlled sizes are used, with the idea that it can probe the critical nucleus of hydrate formation as a function of temperature. The justifiable assumption is that nucleation will occur on the nanoparticle surfaces rather than on a relatively inert glass substrate. The hydrate forming THF-water solution drop was placed in contact with the treated glass surface and monitored during cooling via optical microscopy and DSC, as reported previously for ice nucleation in ref 30. The size and functionality of the tethered particles was controlled for dimensions of approximately 3 – 80 nm (Fig 2a) and hydrate and ice onset and growth attributed to hydrate nucleation at the nanoparticle surfaces. In Figs 2c and S3 we see sharp DSC signals on cooling which are assigned to hydrate and then ice formation, with appropriate melting peaks at 0 and 4.4C. Optical methods are consistent in that the morphology changes at the same temperatures. As we can see in Fig 2a, there is a jump in nucleation temperature (decrease in supercooling) between 31 and 38 nm. For reference, the unit cell of sII hydrates is 1.73 nm. A similar effect is noted for Au/Ag nanoparticles (Fig 4). This is a significant result, obtained by sound sample preparation and experimental methodology.

I found the manuscript addressed an important problem and provided new insight into the process of nucleation. In my view the data show a clear effect with nanoparticle size and functionality and supercooling, and support the conclusions. There is enough detail for the work to be reproduced.

I have two questions

- At a composition of 19 wt % the clathrate should be the only crystal phase seen, but in fact ice is also inferred from DSC. Could this be due, for example, to THF adsorption on the nanoparticles? Can the quantity of ice be estimated?
- GO is soluble in water-THF mixtures, in which case could the experiment be conducted on dissolved

rather than tethered nanoparticles?

Reviewer #4 (Remarks to the Author):

In this work, authors have extended their previous work on ice nucleation (Nature 576, 437-441) to clathrate hydrate nucleation. Experimental techniques and analysis performed is similar to the previous work, only difference being water+THF instead of pure water. While the work is of good quality and useful for clathrate hydrates community, it is a simple extension of their previous work and does not represent a significant advancement in the field as required for publication in Nature Communication. Because of this reason, I recommend that this manuscript be submitted to another journal.

Some thoughts on the paper:

1. Is the surface roughness same across GO and nanoparticles of different sizes. I am curious to see if Wenzel and Cassie-Baxter transition could have any effect on the observed results.
2. With the structure of THF clathrate known, it would be useful if authors can report the number of clathrate cages in the critical nucleus.
3. How long was the system kept at 30C after clathrate melting? Was it long enough so memory effects don't come into picture?
4. Have the authors considered heat transfer on nanoparticles and how could that affect nucleation? Thermal conductivity of solids decreases with decreasing particle size. So, local temperature on a smaller nanoparticle may be different than that on the bigger nanoparticle.

Reviewer #1: This study reports on the nucleation of THF clathrate hydrates on nanosheets and nanocubes. This study actually follows very closely to a prior study published by some of the authors in Nature in 2019 for ice. It is clear that the authors have done extensive work and characterization of the systems considered, which is quite important.

Response: *In this study, we employed a method developed for probing critical ice nuclei using nanoparticles with controlled sizes (Nature 2019, 576, 437-441)¹ to investigate the essential mechanisms behind the THF clathrate hydrate process. In the prior work, a distinct transition in the nanoparticle facilitated ice nucleation was observed, occurring at a consistent value of the nanoparticle size multiplied by the supercooling level across various experimental conditions. This transition was well-explained through the formation of critical ice nuclei, and its size dependence on supercooling was aligned with the principles of classical nucleation theory (CNT). The generality and robustness of these findings, as well as their alignment with theoretical expectations based on simple assumptions, excluded other possible mechanisms linked to nanoparticle-specific effects. This previous work established a robust method for verifying critical ice nucleus formation and assessing its size, an achievement being long sought after due to the transient and nanoscale nature of ice nuclei. This method was also theoretically anticipated to be applicable to various first-order phase transitions, promising to give insights into the key mechanisms of numerous phase transitions.*

Clathrate hydrate nucleation is pivotal in our understanding of phase transitions. However, direct experimental evidence concerning the mechanisms, especially the critical nucleus, remains scarce. While hypotheses exist about the initial stage gas-water cluster formation through molecular simulations and theoretical models; our goal is to address the existence of THF hydrate nucleus, which is a critical gap between the cluster formation and macroscopic hydrates. Our research supplies firstly the experimental evidence of THF hydrate critical nuclei, revealing differences from the CNT - a distinction not found in ice nucleation.

Following the same thoughts as our earlier work on ice nucleation, we employed GO nanosheets, Ag/Au nanocubes of varying sizes, we measured the nucleation temperature of THF hydrates on these diverse nanoparticles, studied how it is correlated with nanoparticle number, examined the average nucleation rate of THF hydrates across different supercooling levels, and analyzed the nucleation temperature distribution of the THF hydrate ensemble. These results collectively validate our conclusions and provide strong support for our findings.

In summary, probing the size of critical nuclei in the formation of THF clathrate hydrates holds significant importance and novelty within the realm of phase transition research for several compelling reasons:

1. Distinct Nature of Ice vs. Clathrate Hydrate Nucleation: *The distinction between ice nucleation and clathrate hydrate nucleation lies in their distinct molecular structures and building units. While ice nucleation has been extensively studied¹, clathrate hydrate nucleation remains almost unexplored due to the complexity of the process. Investigating the size of critical nuclei of THF clathrate hydrates adds a new dimension to our understanding of nucleation in different materials and provides insights into the unique factors governing clathrate hydrate formation.*

2. Unclear Clathrate Hydrate Nucleation Pathway: *The formation of critical nuclei in clathrate hydrates is a topic that lacks conclusive experimental evidence. Understanding the size of critical nuclei in the context of clathrate hydrate nucleation is essential for unraveling the underlying mechanisms. Probing these critical nuclei sheds light on the intricate nucleation process and paves the way for designing strategies to control clathrate hydrate formation.*

3. General Methodology for Nucleation Studies: *The introduction of the developed method to probe critical nuclei, as demonstrated in this study, has broad implications beyond ice or clathrate hydrates. This approach showcases the potential for investigating nucleation processes in various systems, spanning different materials and phase transitions¹⁵. The generality of this method holds promise for advancing our understanding of nucleation phenomena across diverse scientific domains.*

Therefore, probing the critical nuclei of THF clathrate hydrates not only fills a critical knowledge gap specific to clathrate hydrate formation but also contributes to the broader understanding of nucleation processes. It introduces an innovative method applicable to a range of phase transitions, which has the potential to revolutionize how we study and manipulate nucleation in various materials and contexts.

There are several issues with the present manuscript, which diminish the impact and novelty of the study compared to the one published in Nature on ice.

* One of the main concerns here is the choice of clathrate hydrate system. It is understandable that THF was chosen as THF is miscible in water and the formation of clathrate hydrates is a bulk phase transition just as in ice. However, most clathrate hydrates of interest are formed from gas, which are not miscible in water and mass transfer becomes quite significant. For this reason, it is perhaps not surprising that the results reported for THF clathrate hydrates are qualitatively similar to those reported for ice in the Nature paper. As such, the authors should be very explicit that the findings from this study is very specific to THF clathrate hydrates and not for any other type of clathrate hydrates.

Response: *We thank for the reviewer's suggestions and explicitly point the current study is specific to the THF clathrate hydrate. To avoid any misinterpretation, we explicitly acknowledge the limitations and specificity of the findings to THF clathrate hydrates in the revised paper. Additionally, we recommend that future studies explore other clathrate hydrate systems to provide a more comprehensive understanding of nucleation mechanisms across various types of clathrate hydrates formed from different gases.*

1. *Actually, THF-water solutions are very typical as a model clathrate-forming system because they are easily formed as sII clathrate with well-defined quantitative relationship between water and THF molecules at ambient pressure. This system has been extensively studied in previous research to elucidate the characteristic of clathrate hydrates²⁻⁹. When gas molecules are encapsulated by water molecules in gas clathrate hydrates, their behaviors are expected to closely resemble those observed in THF clathrate hydrates.*

2. *Although the mass transfer process in gas hydrates formation maybe different from that of THF hydrates, both gas hydrates and THF hydrates necessarily involve a mixed structure of guest molecules and water molecules in the stage of nucleation, such as the construction of critical nuclei and the further growth of nuclei^{10,11}. For example, some dissolved gas molecules in water participate in the initial construction of the nuclei, and the transferring from neighboring zone and the further dissolving of remaining gas molecules correlates to the further growth of the nuclei. Considering that the size of critical nucleus is found to about a few tens of nanometers normally, as well as the slow growth of the nucleus as an active process before reaching the critical size, the transferring and dissolving of guest molecules in hydrates might more control the growth after the critical nucleus. The current work focuses on the clathrate nucleation of the dissolved molecules due to the much more difficult experimental implementation of the gas hydrates requiring high pressures. It is helpful for understanding gas clathrate which includes both the dissolving, transferring and the nucleating processes. In summary, we believe using the THF system as a model to study the existence and temperature-dependent size of hydrate critical nuclei is justifiable in this context. We also agree with the reviewer's point that in the future it is really important to study the gas hydrates, which is formed from gas immiscible with water.*

* One of the most difficult parts to understand is the results presented in Fig. 2, especially the use of $\Delta T \cdot L$ as an independent variable to quantify the nucleation. This same variable is used in the Nature paper for ice. What is the meaning of this variable and what is so unique about it? What is no unique in going from 31 nm to 38 nm in the size of the GO to cause such change in the nucleation temperature.

Response: *We thank the referee for this comment. Apologies for the early display of the relationship between $\Delta T \times L$ and nucleation temperature. In the previous Nature paper for ice, based on the classical nucleation theory, we expected*

the relationship that the size of critical nucleus was inversely proportional to the supercooled temperature, i.e., $R_c \propto \frac{1}{\Delta T}$; meanwhile, the nucleation efficiency (characterized by the nucleation temperature or the nucleation rate) on finite-size substrates (nanosheets) was discovered to be determined by the dimensionless size of the nanosheets, $\frac{L}{R_c}$. Thus, we expected that the nucleation temperature of THF clathrate was related to $L \cdot \Delta T$ rather than L itself. Thus, in the previous manuscript, we directly plot to show the relation between the experimental nucleation temperature and $\Delta T \times L$ before analyzing more data by varying L and/or ΔT , which were shown in the later figures. We verified the conclusion that the transition can occur on nanosheets with the size $L=31\text{nm}$, 38nm or 46nm in **Figure 3** by varying ΔT while $\Delta T \times L$ reaches a specific value (about 650 nm K) rather than L reaches a specific value between 31 and 38 nm , as shown in **Figure 3** in the main text and SI text. Furthermore, we investigated the general conclusion by using nanocubes. In the revised manuscript, we added more data about the general transition of nucleation efficacy of GOs with various sizes in the new **Figure 2** to make it more readable, and we also added relevant discussion about the detailed explanation of the meaning of $\Delta T \times L$ in the revised manuscript.

* The other variable that is not well defined and explained is tau (= $n \times \text{delay_time}$). From tau, the nucleation rate is obtained. In the text, it is stated that J only depends on T , and not on n . How is this explained? The statements are inconsistent based on the data and equation shown.

Response: We measured the nucleation delay (induction) time of droplets involving n nanosheets (the relative number of the nanosheets as the active sites in facilitating the THF hydrate nucleation) at the temperature T , t_{delay} as a function of both T and n , and estimated the nucleation rate of droplets, $R(T,n) = 1/t_{\text{delay}}(T,n)$. It is expected that R is proportional to the number of nanosheets, n . **Thus the nucleation rate of the THF droplets due to each unit number of nanosheets or nanocubes, $J = R(T,n)/n$, should be a function of only the temperature and independent on the applied n in experiments.** It provides a good verification for the quality of the delay-time measurement. Thus we calculated $\tau = n \times t_{\text{delay}}(T,n)$, which equals to $1/J$ and only a function of temperature T for varying the applied number of nanosheets n . It was well verified by the middle panel of **Figure 3** in main text. We defined explicitly these quantities in the revised main text, and also explained them in **Methods** section (The delay time of THF clathrate nucleation). Our experimental data (**Figure 3**) shows τ is indeed independent of n , which further proves that the nucleation rate is due to nanosheets or nanocubes since $R(T,n)$ is found to be proportional to the number of nanosheets. The measured $t_{\text{delay}}(T, n)$ for various T and n collapsed to form the single curve $\tau(T) = n \times t_{\text{delay}}(T, n)$ within a larger T range which is further identify the transition of the curve $\tau(T)$, or the $J(T)=1/\tau(T)$, more clearly.

* Another major concern in the results is that the DSC scans show the formation of ice. At the THF concentration used, which is the stoichiometric concentration for forming structure II clathrate hydrates, no water should be available to form ice. THF is of course volatile and should evaporate during the experiment. However, how can one be confident that the phase transition measured is actually that for THF clathrate hydrates and not ice?

Response: We have refined the expression to show our experiment. To address this concern and enhance the clarity of our findings, we firstly conducted additional experimental loops in the DSC measurements to ensure that THF did not evaporate out of the sealed cell during the experiments. Secondly, we compared our DSC results with existing literature on both THF clathrate hydrates and ice formation. This comparative analysis allows us to contextualize our findings within the broader scientific landscape, distinguishing between the characteristics of THF clathrate hydrates

and ice formation under similar conditions. As shown in **Figure R1**, we carried out DSC scans of the same sample with two loops, in which the sample was kept at 30 °C for 30 minutes (the formed ice and hydrates are found to have completely melted) after the first-cycle scan, and then commencing the testing for the second-cycle scan. As shown in the inset, formed ice was distinguished by analyzing the first melting peak, which corresponds to a melting point of approximately 0 °C. In contrast, THF was identified by the second melting peak, with a melting point of approximately 4.4 °C. This clear differentiation between the melting points of ice and THF provides a reliable basis for discerning and characterizing the observed phase transitions accurately. From the two loops, the formed ice (area of ice peaks) in the two loops are almost the same with the difference of 0.2%, and the formed THF hydrate are also almost the same with the difference of 1.5% (the red and black lines in the inset, respectively), indicating the excellent sealing in DSC scans. Actually, in a sealed state, the occurrence of both THF hydrate and ice crystallization peaks in DSC during the crystallization process of the THF and water mixture is related to the design of the experimental setup, rather than indicating a leakage of gas^{9,12,13}.

Figure R1. DSC scans with two loops and excellent sealing

Moreover, as shown in **Figure R2**, from the integrated area (A) of ice peaks in DSC graphic, according to the melting heat $Q = KA$ ($K = 0.9$ for the DSC8500 machine) and ice melting heat value at 0 °C is water-ice heat of phase transformation = 334 J/g, the total water is $\text{Volume} \times \text{Density}$. The formation of average ice content during the cooling

cycle can be calculated as $\text{ice mass} = A \times 0.9 \times 10^{-3} / 334$ in gram and the results are listed in the **Table R1**. It shows that ice can be formed while the THF/water ratio of the solution is between 15% and 23%, which reveals that even the slight excess of THF (the ratio can be larger than that in hydrates 19%) can still produce ice in cooling the THF aqueous solution, and only when the THF/water ratio is higher (e.g. 28%), there is not water to form ice.

Figure R2. Ice formation during cooling the THF solution with various THF/water ratio

THF/H ₂ O ratio (wt)	15% THF + 85% H ₂ O	19% THF + 81% H ₂ O	23% THF + 77% H ₂ O	28% THF + 72% H ₂ O
H ₂ O total mass	0.00255g	0.00240g	0.00231g	0.00216g
Ice mass	0.00026g	0.00018g	0.00015g	0g
Ice mass/H ₂ O total mass	10.2%	7.5%	6.5%	0%

Table R1. Calculated ice content by cooling the THF aqueous solutions with varying THF contents.

In addition, before and after conducting every nucleation experiment, the sample sealed in the cell was weighed, and the difference was within 0.02mg (0.7% of the total weight of solution), ensuring that the accuracy of the experimental results would not be affected by losses of THF and water during the experimental time. In cases where the cell is not hermetically sealed, both water and THF are susceptible to evaporate at a temperature of 30 °C within a span of 30 minutes. Under such circumstances, the disparity between the two weight measurements would exceed 0.02mg. This observation underscores the importance of maintaining a proper seal to ensure the accuracy and reliability of the experimental results. Under the optical microscopy, the formation of THF clathrate as transparent, needle-like crystal and ice as black and block-like substance with sudden change of the opacity as shown in **Figure 1** in main text. When both two crystalline phenomena appeared, the corresponding THF hydrate and ice nucleation temperature were recorded.

Moreover, in our experimental setup, the liquid THF-water mixture being tested did not completely fill the container (the sealed cell), and there was gas space above the liquid surface and below the container lid. As a result, some water vapor and THF gas evaporated from the liquid. The saturated vapor pressure of THF is higher than that of water, the proportion of THF molecules in the evaporated gas mixture is higher than that of water molecules (especially, larger than the 1/17 of evaporated water molecules). Thus, a small amount of water molecules in the liquid will be exceeded in the formation of THF hydrates, which may be the reason of forming ice.

* The data in Fig. 3 for the 38 nm GOs does not show the clear breaks for the delay time and tau, which in panel b, surprisingly shows a clear separation of the data above and below the subcooling of 17 K.

Response: As shown in **Figure R3**, due to the limited measured range of delay time, for each n (relative number of nanosheets as nucleation sites), the delay time of nucleation can be achieved only in a small temperature range, thus a single delay time-temperature curve for a specific n , i.e., $t_{\text{delay}}(T; n)$, maybe not very clearly show the rapid change over the specific temperature (for example, it may mainly locate at one side of the transition temperature rather than well cross it for identifying the transition). The shorter curve as well as the error bar might hinder to identify the transition very clearly, as shown in the panel a. Thus, we test $n \cdot t_{\text{delay}}(T, n)$ and verify that the curves from different n collapse to the almost same curve within errors as the expectation, thus the formed single curve $\tau(T) = n \cdot t_{\text{delay}}(T, n)$ covers a large temperature range crossing over the transition temperature for easily identifying the (derivative) discontinuous of the nucleation rate. The data shown in the panel b (J) is to directly transfer from that in the panel a ($J \sim I = t_{\text{del}}$) without any additional treatment except adding the guiding straight lines. Then, the transition is much easily to be identified.

Figure R3. Transitions in the nucleation rate during the THF clathrate formation GO nanosheets

Minor points:

* Units for the variables should be consistent. For example, some temperatures are in degree C and others in K.

Response: Thanks. We have corrected all of the unit as °C in the revised manuscript.

* The details of what is DG* are said to be in the Methods section, but they are no where to be found.

Response: Thanks for your careful reading, and apologies for the missing details of ΔG^* . We have added the discussion in the main text and **Figure 4** in the revised manuscript.

* Some parts of the text needs improvement.

Response: Thanks. We followed your advice to refine our expressions in the revised manuscript.

* The title is misleading and should be more specific to the system and conditions studied.

Response: Thanks. We have updated the title of “probe critical nuclei size of THF clathrate hydrate” to specifically focus on the THF system.

Reviewer #2: In this manuscript, the authors report the nucleation and crystallization of THF clathrate hydrate. GO nanosheets with certain sizes can reduce the degree of supercooling in the THF clathrate hydrate formation. The reported data are quite interesting and they would give us new knowledge in the nucleation of THF clathrate hydrate. The methodology is also interesting. However, the reviewer feels the present manuscript needs major revisions before publication.

1. The most significant point the authors need revising is that the authors have never observed the existence of critical nucleus directly. The authors changed the size of GO nanosheets and measured the induction time and the degree of supercooling in the THF clathrate hydrate formation. The authors should clarify why the authors can discuss the size of the critical nucleus on the basis of GO nanosheet experiments. Generally, the size of the critical nucleus should be smaller than 10 nm. The reviewer thinks the experimental results do not always support the conclusions.

Response: *Thanks for the comments. We have discussed the details in the main text (Theoretical calculation of free-energy barrier of THF clathrate nucleation on finite-sized substrates) in the revised manuscript. The nucleation free energy barrier (thus the nucleation occurrence temperature at cooling experiments or the nucleation delay (induction) time at constant temperature experiments) on finite-sized substrates is dependent on the relative value of the size of nanoparticles to the diameter of critical nucleus, $l = \frac{L}{2R_c}$. We have the nucleation free energy barrier on the nanoparticles $\Delta G^*(L; \Delta T) = \Delta G_{het}^*(\Delta T) \hat{g}(l)$. Here ΔG_{het}^* is the normal heterogeneous nucleation barrier on the same material but with infinite size, and $\hat{g}(l)$ is unit while l is larger than a specific constant value $l_c \sim 1$ and starts to quickly increase while $l < l_c$. The function $\hat{g}(l)$ is approximately general for various particles, which is not sensitive to physical properties of particles (nanosheets or nanocubes) and can be estimated by the classical nucleation theory or simulations. Therefore, the nucleation temperature (or nucleation rate) will have a rapid change while the size of particles L equals to a specific value $L_c = 2R_c l_c$, and we can get the size of critical nucleus $R_c = L_c / 2l_c$ from the measured L_c . In this work, we generally find that the THF hydrate nucleation on nanosheets (and nanocubes) has a transition at a specific value of $L \Delta T \approx c$, the product of the size of nanosheets, L , and the supercooled temperature of the nucleation in a variety of conditions (on GO nanosheets with 31, 38, and 46nm in size, respectively; on Ag, or Au nanocubes with various sizes, respectively; in cooling, and constant temperature experiments), thus we have the size of critical nucleus, $R_c = \frac{c}{2l_c} \frac{1}{\Delta T} \approx \frac{300}{\Delta T}$ nm, which is inversely proportional to the supercooled temperature ΔT , being well agreement with the expectation of classical nucleation theory. Therefore, we actually present a nanometer ruler (nanosheets or nanocubes with a specific size) and detect the nucleation signal on it. While the size of critical nucleus (dependent on ΔT) matches that of the nanometer ruler, the clathrate forms thus the size of critical nucleus can be measured. This work provides a general and powerful method to measure the size of the small and transient critical nucleus (there does not exist another method to measure so far) as well as its temperature dependence. It is worth mentioning, while the radius of critical nucleus for ice nucleation, $R_c \approx \frac{100K}{\Delta T}$ nm, thus about 10nm in the normal supercooled temperature, $\Delta T = 10K$ (in previous paper, Nature **2019**, 576, 437-441), this current work about THF clathrate nucleation gives an about three-times larger size of critical nucleus than that of ice surprisingly, which has never been reported.*

2. Recently, Machida et al (CrystEngComm, vol. 24, pp. 6730 (2022)) reported the different nucleation mechanism of THF clathrate hydrate. They stated the mechanism is not the classical nucleation theory, but similar to labile cluster or blob mechanism. Though the results are different from the ones the authors suggested, there seems to be similar points on the sizes and/or the interaction between THF solution and materials (GO sheets or crystal seeds). At least, the author should discuss their results in the manuscript.

Response: *Thanks for the comment, and we have discussed the reference in the revised manuscript. In that reference, the author found the addition of AgO or Ag₃PO₄ (average size was 1 to 100 μm) effectively diminishing the degree of*

supercooling in the THF hydrate formation. Furthermore, in the THF aqueous solution without AgO, the 10–30 nm clusters are almost non-existent at 280 K and 278 K. It was very difficult to find a number of clusters, although each SEM image included several. At 258 K, which is just before crystallization at 253 K, a number of 10–30 nm clusters and their aggregates were observed¹⁴.

The Ag-containing nano-protrusions on the substrate surface promote the nucleation of hydrates, and the nucleation structure can be observed under conditions around 20 K supercooling, forming 10-30 nm nano-clusters. According to our theory, the minimum stable critical nucleus diameter generated under 20 K supercooling should be 30 nm. This result actually confirms the applicability and correctness of our study and we have cited that paper.

3. The results clearly indicate the product of the degree of supercooling and the size of GOs would be an index as THF clathrate hydrate nucleation. However, the reviewer hardly understand its scientific meanings. Moreover, when Ag (or Au) nanocubes were used instead of GOs, the index was also changed. The reviewer wants the authors to discuss the scientific meanings.

Response: As we answered in question 1, the nucleation free energy barrier on finite-size nanoparticles has a transition while the relative size of nanosheets (or nanocubes) to the diameter of critical nucleus, $l = L/2R_c$, equals to a specific value l_c , and the l_c is in the order of unit, which is not very sensitive to the detailed physical properties of nanosheets (and nanocubes), but is slightly dependent on the shape of nanosheets, varying from $l_c \approx 1$ for the circular nanosheets, to $l_c \approx 0.8$ for the cubic nanocubes. The conclusion can be obtained based on the spirit of classical nucleation theory. The details about the transition of nucleation at l_c as well as the value of l_c on different shape of nanoparticles are shown in the section of supplementary. In addition, based on the spirit of classical nucleation theory, the radius of critical nucleus is (approximately) inversely proportional to the supercooled temperature, i.e., $R_c = \frac{c}{\Delta T}$, with the approximated constant c . Therefore, we have that the nucleation transition generally occurs while $L \approx 2l_c R_c = 2cl_c/\Delta T$, i.e., $L\Delta T \approx 2cl_c$. This is the central result found in this work. We generally found the relation for the THF nucleation on GO nanosheets ($2cl_c \approx 650 \text{ nm K}$, for $l_c \approx 1$, thus the estimated radius of THF critical nucleus on the GO nanosheets $R_c \approx \frac{325}{\Delta T} \text{ nm}$) and on Ag/Au nanocubes ($2cl_c \approx 450 \text{ nm K}$ for $l_c \approx 0.8$, thus the estimated radius of THF critical nucleus on the nanocubes $R_c \approx \frac{280}{\Delta T} \text{ nm}$). The relative difference in GOs and in Au/Ag is estimated about 10%. Considering that the uncertainties in nanosheet (and nanocubes) size and shape, we estimate the radius of critical nucleus of THF about $\frac{300}{\Delta T} \text{ nm}$ with 10-20% error.

4. The authors should cite other recent articles on THF clathrate hydrate nucleation.

For example, Huang et al., J.Phys.Chem.C, vol. 124, pp. 13966 (2020) and Cryst. Growth Des., vol. 20, pp. 2921 (2020), and Ghosh et al., J.Phys.Chem.C, vol. 123, pp. 16300 (2019).

Response: Thanks for the recommended references. We have cited and discussed the articles in the revised manuscript.

Reviewer #3: This manuscript investigates the formation of THF-water clathrate hydrate in the presence of nanoparticle coated surfaces. THF-water solutions are very topical as a model clathrate-forming system, as THF and water mix at the required composition and the crystallisation temperature is around 277K at ambient pressure. This contrasts for example with methane-hydrates, where nucleation and growth are limited by the gas solubility and surface area of the gas-liquid interface.

Here, surface-anchored graphene oxide (GO) and Au/Ag nanoparticles with controlled sizes are used, with the idea that it can probe the critical nucleus of hydrate formation as a function of temperature. The justifiable assumption is that

nucleation will occur on the nanoparticle surfaces rather than on a relatively inert glass substrate. The hydrate forming THF-water solution drop was placed in contact with the treated glass surface and monitored during cooling via optical microscopy and DSC, as reported previously for ice nucleation in ref 30. The size and functionality of the tethered particles was controlled for dimensions of approximately 3 – 80 nm (Fig 2a) and hydrate and ice onset and growth attributed to hydrate nucleation at the nanoparticle surfaces. In Figs 2c and S3 we see sharp DSC signals on cooling which are assigned to hydrate and then ice formation, with appropriate melting peaks at 0 and 4.4C. Optical methods are consistent in that the morphology changes at the same temperatures. As we can see in Fig 2a, there is a jump in nucleation temperature (decrease in supercooling) between 31 and 38 nm. For reference, the unit cell of sII hydrates is 1.73 nm. A similar effect is noted for Au/Ag nanoparticles (Fig 4). This is a significant result, obtained by sound sample preparation and experimental methodology.

I found the manuscript addressed an important problem and provided new insight into the process of nucleation. In my view the data show a clear effect with nanoparticle size and functionality and supercooling, and support the conclusions. There is enough detail for the work to be reproduced.

Response: *We appreciate that this referee reviewed this work very favorably.*

I have two questions

· At a composition of 19 wt % the clathrate should be the only crystal phase seen, but in fact ice is also inferred from DSC. Could this be due, for example, to THF adsorption on the nanoparticles? Can the quantity of ice be estimated?

Response: *We thank the referee for this comment. In a sealed state, the occurrence of both THF hydrate and ice crystallization peaks in DSC during the crystallization process of the THF and water mixture is related to the design of the experimental setup, rather than indicating a leakage of gas^{9,12,13}. We test the dependence of the quantity of formed ice on the concentration of THF in the solution. As shown in **Figure R2** and Table R1, the quantity of ice decreases as increasing the concentration, but reach zero until the concentration is a larger value than the expected 19%, (e.g., 28%), consisting with the argument that the remained water molecules inside the liquid droplet are excessed after the hydrate due to the evaporation of water molecules to the gas space less than the 17 times (the value in the clathrate hydrates) of that of THF molecules.*

*In our experimental setup, the liquid THF-water mixture being tested did not completely fill the container, and there was gas space above the liquid surface and below the container lid. As a result, some water vapor and THF gas evaporated from the liquid. Since the saturated vapor pressure of THF is higher than that of water, the proportion of THF molecules in the small amount of gas mixture is higher than that of water molecules. Thus, a small amount of water molecules in the liquid will not participate in the formation of THF hydrates. In addition, before and after conducting every nucleation experiment, the sample was weighed twice, and the difference between the two measurements was within 0.02mg (about 0.7% of the total weight of the applied THF solution), ensuring that the accuracy of the experimental results would not be affected by losses (if having) of THF during the experimental time. As shown in **Figure R1**, From the integrated area (A) of ice peaks, according to the melting heat $Q = KA$, ($K = 0.9$ for the DSC8500 machine) and ice melting heat value at 0°C is water-ice heat of phase transformation = 334 J/g. The total water is $3 \times 0.81 \times 10^{-3} = 0.0024\text{g}$. The formation of averaged ice content during the cooling cycle can be calculated as $((64.29 + 64.27) / 2) \times 0.9 \times 10^{-3} / 334 = 0.00017\text{g}$. It indicates that 5.8% water (comparing to the solution in weight) forming the ice, which not forming the THF clathrate hydrate.*

· GO is soluble in water-THF mixtures, in which case could the experiment be conducted on dissolved rather than tethered nanoparticles?

Response: *Thank you for your comment. We incorporate the following explanation into the main text of the revised manuscript. It is possible to do the similar experiments by dissolving GO in the water-THF mixtures, instead of using*

the tethered nanoparticles. In our previous work about ice nucleation, we dissolved GOs in water as well as tethered GOs on surface and we got the same results. However, due to the larger size of the applied GOs in this work, they may deviate from flat sheets, overlap and aggregate partially, and move, if dissolving them in the mixtures. It would be much more difficult to control the size and concentration of the nanoparticles, as they would be more prone to aggregation and precipitation. Additionally, the use of tethered nanoparticles allows for a more precise determination of the density (D) and number (n) of the nuclei, and controls of the nucleation to occur at surface. Therefore, in this work, we only tethered nanoparticles on surfaces.

Reviewer #4: In this work, authors have extended their previous work on ice nucleation (Nature 576, 437-441) to clathrate hydrate nucleation. Experimental techniques and analysis performed is similar to the previous work, only difference being water+THF instead of pure water. While the work is of good quality and useful for clathrate hydrates community, it is a simple extension of their previous work and does not represent a significant advancement in the field as required for publication in Nature Communication. Because of this reason, I recommend that this manuscript be submitted to another journal.

Response: *The Nature paper previously published by the research team presents an effective strategy for detecting the critical nucleus in ice crystallization and it was expected to be a general strategy for various the first order phase transition. Actually, as the unique strategy to experimentally investigate the small and transient critical nucleus, it is important to apply and extend the method to detect the nucleation mechanism of various nucleation processes including the clathrate hydrates, and get insights of the nucleation by verifying the existence and properties of the critical nucleus*¹⁵.

It is noted that the nucleation mechanism of clathrate hydrates is still not clear. The main focus of this research is to investigate the mechanism of the THF clathrate hydrate nucleation and gives insights about the clathrate hydrate nucleation by answering experimentally some important (and unknown) questions: Whether the THF nucleation is controlled by the formation of critical nucleus? What are the size and its temperature dependence of the THF critical nucleus? How much is the (nanometer-scale) THF clathrate crystalline critical nucleus different from its macroscopic counterpart? So far, it is very lacked to achieve answers of these key questions by experiments (and even by simulations). Therefore, it is not a simply extending of the previous work¹, but a significant study for understanding clathrate formation, as well as extends and verifies the powerful strategy.

In the revised manuscript, we emphasize the scientific significance of the verification of the existence of critical nuclei and the detection of the critical nucleus size in THF clathrate formation. The critical nucleus as the central concept and the control step for the clathrate hydrate formation has never been detected in experiments. This study clearly reveals experimentally the key step during the THF clathrate hydrate nucleation, which can be extended for providing insights about clathrate hydrates.

1. Is the surface roughness same across GO and nanoparticles of different sizes. I am curious to see if Wenzel and Cassie-Baxter transition could have any effect on the observed results.

Response: *The nanoparticles of different types with various sizes are possible to have different sub-nanometer structure and properties, involving different subnanometer surface roughness. However, we found that the nucleation efficiency of various nanoparticles with different sizes has a general transition at the size-inverse-proportional supercooling. It actually proves that the observed results are not due to the details of nanoparticles, such as the subnanometer surface roughness, while it is explained very well based on the existence of critical nuclei.*

*Our experimental results showed (seeing **Figure 4** in main text), the THF clathrate nucleation occurs on the GO*

nanosheets (or Ag/Au nanocubes) with the size about tens of nanometers as nucleation sites rather on the glass substrate. The size of these nanosheets (or nanocubes) is in tens of nanometers, the distance between neighboring tethered nanoparticles is also in this scale, thus the roughness of the glass surface tethered with various nanoparticles does not bring the Wenzel and Cassie-Baxter transition of the THF clathrate resolution on the surface, which requires the structure in micrometer scale and low-surface-tension surface. In addition, the ultra-structural surface details of nanosheets may averagely affect the THF clathrate nucleation on them, but it is hard to imagine that it can bring the transition of nucleation generally observed at a specific value of the product of the size of nanosheets and the supercooled temperature. As a contrast, the generally found transition of nucleation is easy to explain from the dependence of the nucleation free energy barrier on the relative size of nanosheets to critical nucleus.

We also checked the roughness of single layer GO nanosheets and the metal nanocubes by using atomic force microscopy (AFM). The results showed that GO (38 nm) surface roughness is 0.41 ± 0.01 nm, the Au nanocube (45 nm) surface roughness is 0.53 ± 0.01 nm, and Ag nanocube (45 nm) roughness is 0.51 ± 0.01 nm. The roughness of the surfaces changed little. Therefore, we inferred that the subnanometer surface roughness has little effect on the general occurred transition of clathrate nucleation on the specific value of the product of nanoparticle size and supercooling.

2. With the structure of THF clathrate known, it would be useful if authors can report the number of clathrate cages in the critical nucleus.

Response: In the example of a nucleus of THF clathrate hydrate on the GOs with a length size of 38 nm and an area of 361 nm^2 , we have that the nucleus accommodate a maximum of 208 unit cells on its contact layer with GOs, by using the lattice parameter of THF sII clathrate hydrate crystal, $a=1.731 \text{ nm}$ (Face-centered cubic Fd-3m). If we assume a homogeneous distribution of sII unit cells, an estimation can be made. The number of 5^{12} cage would be 16 times the maximum number of unit cells (208), resulting in $16 \times 208 = 3328$ cages. Similarly, the number of $5^{12}6^4$ cage would be 8 times the maximum number of unit cells, resulting in $8 \times 208 = 1664$ cages.

We consider the contact angle between critical nuclei and substrate, for example, a possible value about 30 degrees. Thus, as illustrated in **Figure R4**, $r_s = r_c / \cos 30 = 38 / 0.866 = 43.9 \text{ nm}$, $h = r_s - r_c \times \tan 30 = 43.9 - 21.9 = 22 \text{ nm}$, the volume of the critical nuclei is $\pi \times h \times (3 \times r_c^2 + h^2) / 6 = 3.14 \times 22 \times (3 \times 38^2 + 22^2) / 6 = 55448 \text{ nm}^3$. The critical nucleus could accommodate a maximum of $55448 / 1.731^3 = 10690$ unit cells.

Please note that this estimation is based on assumptions and the actual structure of the critical nucleus may vary. Further refinement and experimental analysis would be required to determine the accurate cage type and cage numbers in the critical nucleus.

Figure R4. Volume calculation of cap model of critical nuclei

3. How long was the system kept at 30C after clathrate melting? Was it long enough so memory effects don't come into picture?

*Response: We held the melted THF clathrate at 30 °C for 30 minutes for eliminating the memory effect according to previous discoveries as shown in reference⁸¹⁶. Moreover, as shown in the **Figure RI**, for showing the elimination of memory effect, we carried out DSC scans of the same sample with two loops, in which the sample was kept at 30 °C for 30 minutes after the first-cycle scan, and then commencing the testing for the second-cycle scan. The results showed that the THF clathrate hydrate nucleation temperature in the second cycle is not increased. Thus, it indicates that our experimental setup of holding the sample at 30 °C for 30 minutes can eliminate the memory effect in the nucleation tests.*

4. Have the authors considered heat transfer on nanoparticles and how could that affect nucleation? Thermal conductivity of solids decreases with decreasing particle size. So, local temperature on a smaller nanoparticle may be different than that on the bigger nanoparticle.

Response: We took this into consideration by using the slower cooling rates to reduce the possible temperature inhomogeneity in our experiment setup thus the possible impact on nucleation. Additionally, the delay time testing is more less affected by the possible different thermal conductivity of the experiment setup on nucleation.

The heat transfer rate (Q) is proportional to the temperature difference (ΔT), the area (A), and the thermal conductivity (k) of the material, while inversely proportional to the distance or thickness (d) between the surfaces.

$$Q = \Delta T \times k \times A / d$$

For the cooling rate of 1 K/min, the $Q = 0.0003$ J/s for the Linkam LTS420.

For the graphene oxide $k_{GO}=72 \text{ Wm}^{-1}\text{K}^{-1}$, thickness $d=0.8 \text{ nm}$, area $A= (38/2)^2\pi \text{ (nm}^2\text{)}$

$$\Delta T_{GO} = (0.8 \times Q)/(72 \times 361\pi) = 9.6Q \text{ (K)} = 0.0035 \text{ K}$$

silver nanocube $k_{Ag}=420 \text{ Wm}^{-1}\text{K}^{-1}$, thickness $d=45 \text{ nm}$, area $A= (45/2)^2\pi \text{ (nm}^2\text{)}$

$$\Delta T_{Ag} = (45 \times Q)/(420 \times 506.25\pi) = 67Q \text{ (K)} = 0.021 \text{ K}$$

gold nanocube, $k_{Au}=318 \text{ Wm}^{-1}\text{K}^{-1}$, thickness $d=45 \text{ nm}$, area $A= (45/2)^2\pi \text{ (nm}^2\text{)}$

$$\Delta T_{Au} = (45 \times Q)/(318 \times 506.25\pi) = 89Q \text{ (K)} = 0.028 \text{ K}$$

gold nanocube, $k_{Au}=318 \text{ Wm}^{-1}\text{K}^{-1}$, thickness $d=70 \text{ nm}$, area $A= (70/2)^2\pi \text{ (nm}^2\text{)}$

$$\Delta T_{Au} = (70 \times Q)/(318 \times 1225\pi) = 57Q \text{ (K)} = 0.019 \text{ K}$$

In general, thermal conductivity may have a significant impact on crystal growth rate. During crystal growth, the large amount of phase-transition heat is transferred through thermal conduction from the local high-temperature region to the low-temperature region, causing a change in the crystal growth rate due to the different local temperatures. It might be possible that the different-sized nanoparticles affect the local temperature nearby the nanoparticles thus the growth rate of crystals locally. However, the impact of thermal conductivity on nucleation is much smaller because the temperature change during the formation of critical nuclei is small, and the heat does not need to conduct completely from the nanoparticles, so that the possibly different thermal conductivity of these slightly different size nanoparticles less influences the local temperature during nucleation^{17,18}. It is not responsible to the generally found transition at the specific value of the product of the size of nanoparticles and supercooling.

Reference

- 1 Bai, G., Gao, D., Liu, Z., Zhou, X. & Wang, J. Probing the critical nucleus size for ice formation with graphene oxide nanosheets. *Nature* **576**, 437-441, doi:10.1038/s41586-019-1827-6 (2019).
- 2 Devarakonda, S., Groysman, A. & Myerson, A. S. THF-water hydrate crystallization: an experimental investigation. *J Cryst Growth* **204**, 525-538, doi:Doi 10.1016/S0022-0248(99)00220-1 (1999).
- 3 Lehmkuhler, F. *et al.* Temperature-Induced Structural Changes of Tetrahydrofuran Clathrate and of the Liquid Water/Tetrahydrofuran Mixture. *J Phys Chem C* **115**, 21009-21015, doi:10.1021/jp207027p (2011).
- 4 Liu, Z., Kim, J., Lei, L., Ning, F. & Dai, S. Tetrahydrofuran Hydrate in Clayey Sediments—Laboratory Formation, Morphology, and Wave Characterization. *Journal of Geophysical Research: Solid Earth* **124**, 3307-3319, doi:10.1029/2018jb017156 (2019).
- 5 Makogon, T. Y., Larsen, R., Knight, C. A. & Sloan, E. D. Melt growth of tetrahydrofuran clathrate hydrate and its inhibition: method and first results. *J Cryst Growth* **179**, 258-262, doi:Doi 10.1016/S0022-0248(97)00118-8 (1997).
- 6 Wu, J. Y., Chen, L. J., Chen, Y. P. & Lin, S. T. Molecular Dynamics Study on the Equilibrium and Kinetic Properties of Tetrahydrofuran Clathrate Hydrates. *J Phys Chem C* **119**, 1400-1409, doi:10.1021/jp5096536 (2015).
- 7 Yang, M. *et al.* Effects of Additive Mixture (THF/SDS) on the Thermodynamic and Kinetic Properties of CO₂/H₂ Hydrate in Porous Media. *Ind Eng Chem Res* **52**, 4911-4918, doi:10.1021/ie303280e (2013).
- 8 Zeng, H., Wilson, L. D., Walker, V. K. & Ripmeester, J. A. Effect of antifreeze proteins on the nucleation, growth, and the memory effect during tetrahydrofuran clathrate hydrate formation. *J Am Chem Soc* **128**, 2844-2850, doi:10.1021/ja0548182 (2006).
- 9 Zhang, Y., Debenedetti, P. G., Prud'homme, R. K. & Pethica, B. A. Differential Scanning Calorimetry Studies of Clathrate Hydrate Formation. *The Journal of Physical Chemistry B* **108**, 16717-16722, doi:10.1021/jp047421d (2004).
- 10 Jacobson, L. C. & Molinero, V. Can amorphous nuclei grow crystalline clathrates? The size and crystallinity of critical clathrate nuclei. *J Am Chem Soc* **133**, 6458-6463, doi:10.1021/ja201403q (2011).
- 11 Warriar, P., Khan, M. N., Srivastava, V., Maupin, C. M. & Koh, C. A. Overview: Nucleation of clathrate hydrates. *J Chem Phys* **145**, 211705, doi:10.1063/1.4968590 (2016).
- 12 Arzbacher, S., Petrasch, J., Ostermann, A. & Loerting, T. Micro-Tomographic Investigation of Ice and Clathrate Formation and Decomposition under Thermodynamic Monitoring. *Materials (Basel)* **9**, doi:10.3390/ma9080668 (2016).
- 13 Kumar, A., Kumar, R. & Linga, P. Sodium Dodecyl Sulfate Preferentially Promotes Enclathration of Methane in Mixed Methane-Tetrahydrofuran Hydrates. *iScience* **14**, 136-146, doi:10.1016/j.isci.2019.03.020 (2019).
- 14 Machida, H., Sugahara, T. & Hirasawa, I. Supercooling suppression in the tetrahydrofuran clathrate hydrate formation. *CrystEngComm* **24**, 6730-6738, doi:10.1039/d2ce00645f (2022).
- 15 Jin, L., Shi, Y., Allen, F. I., Chen, L. Q. & Wu, J. Probing the Critical Nucleus Size in the Metal-Insulator Phase Transition of VO₂. *Phys Rev Lett* **129**, 245701, doi:10.1103/PhysRevLett.129.245701 (2022).
- 16 Wilson, P. W. & Haymet, A. D. J. Hydrate formation and re-formation in nucleating THF/water mixtures show no evidence to support a “memory” effect. *Chemical Engineering Journal* **161**, 146-150, doi:10.1016/j.cej.2010.04.047 (2010).
- 17 Vekilov, P. G. Nucleation. *Cryst Growth Des* **10**, 5007-5019, doi:10.1021/cg1011633 (2010).
- 18 Li, K. *et al.* Investigating the effects of solid surfaces on ice nucleation. *Langmuir* **28**, 10749-10754, doi:10.1021/la3014915 (2012).

REVIEWER COMMENTS

Reviewer #1 (Remarks to the Author):

The authors have put a good amount of effort in answering the reviewers' concerns. However, fundamentally there is still two issues that are difficult to reconcile toward favoring the publication of this work in Nat Comm.

* This work is clearly an extension of the previous work on ice published in Nature. The methods, analysis, and discussion follow a very similarly to the previous paper.

* The authors insist to claim that the nucleation of THF clathrate hydrates is representative of clathrate hydrates of gas, which is just incorrect and worlds apart. The nucleation of THF clathrate hydrates is indeed similar to ice as it is a bulk nucleation phenomena, whereas, as pointed in the original review, for clathrate hydrates of gas, the process is severely mass transfer limited, which is not presented for THF. The vast majority of interest in clathrate hydrates is on gas, not THF. As such, there is limited translation of these learnings to system of actual interest. This does not question the scientific effort and gain from the work performed, but its impact and important is limited.

Reviewer #2 (Remarks to the Author):

At a glance, the authors addressed some points the reviewer suggested previously. But. in the text, the discussion on the previously-reported nucleation mechanism of THF clathrate hydrate was insufficient. The authors just added the references in the text. They only discussed in the responses to reviewers. According to the reports by Machida et al., they observed the cluster in the THF aqueous solution by SEM before nucleation. They reported the size of clusters and the temperature dependence of number density of clusters. Probably the clusters in the literature would be equal to the critical nucleus the authors. Why the authors do not compare their results with the SEM observation results? The reviewer believes the addition of the quantitative discussion of the size makes this manuscript better. It should lead to further scientific understanding of the product of ΔT and L .

Reviewer #3 (Remarks to the Author):

In my original report I found the article to be of general interest and technically sound. I raised two questions which I feel have been satisfactorily addressed in the response.

"At a composition of 19 wt % the clathrate should be the only crystal phase seen, but in fact ice is also inferred from

DSC. Could this be due, for example, to THF adsorption on the nanoparticles? Can the quantity of ice be estimated?". The authors have responded to this point by clarifying the experimental method and possible origins of pure ice formation. This probably doesn't rule out THF absorption on the nanoparticles, but the in my view the conclusions are robust in any case.

"GO is soluble in water-THF mixtures, in which case could the experiment be conducted on dissolved rather than

tethered nanoparticles?". The authors respond by explaining why solution studies might be more complicated than in pure water/ice, for example by GO folding, and have added to the main text.

Some minor comments/suggestions on the highlighted text.

Title, I suggest: "Probing the critical nucleus size in tetrahydrofuran clathrate hydrates"

Line 92: "The remaining"

Line 264: "The dependence of the free energy barrier" (remove "In physics").

Reviewer #4 (Remarks to the Author):

The effort taken by the authors to thoroughly respond to all reviewer comments is indeed commendable. I believe the manuscript still needs revision before it can be accepted for publication.

1. The critical nucleus size and number of cages reported in this study are significantly larger than what's reported in literature for clathrate hydrates. Authors should use their results to prove/disprove available MD simulation/experimental results on clathrate hydrate critical nucleus size. For example, Jacobson and Molinero (<https://pubs.acs.org/doi/pdf/10.1021/ja201403q>) report critical nucleus size of 3-4 nm. They also report subcooling vs. nucleus size from MD simulations. Does that agree with $R_c=300/dT$ found in this study? If not, why? It will add value to the manuscript if authors can compare their results with literature values and provide a discussion on why they think their results are different.

2. Regarding thermal conductivity of nanoparticles, please see following works on size effects:

<https://pubs.acs.org/doi/10.1021/nl1045395>

<https://journals.aps.org/prb/abstract/10.1103/PhysRevB.61.2651>

In their analysis, authors assumed same thermal conductivity for different sized particles. In fact, thermal conductivity of smaller nanoparticles in their work can be 1-2 order of magnitude lower than bulk value. The thermal analysis is not straight forward as authors assumed.

Reviewer #1 (Remarks to the Author):

The authors have put a good amount of effort in answering the reviewers' concerns. However, fundamentally there is still two issues that are difficult to reconcile toward favoring the publication of this work in Nat Comm.

* This work is clearly an extension of the previous work on ice published in Nature. The methods, analysis, and discussion follow a very similarly to the previous paper.

***Response:** The methodology for probing the size of critical nuclei, utilizing nanoparticles with controlled sizes, was initially designed for the ice nucleation processes. The theoretical expectation was that this method could be applicable to a wide range of nucleation processes associated with the first-order phase transitions, offering a general and powerful tool to investigate the formation of critical nuclei, which is difficult to tackle with existing techniques^{1,2}. Therefore, practical implementation and optimization of this methodology to various nucleation processes are crucial. Such endeavors are essential not only to verify the robustness and generality of the method but also to uncover possible limitations. Most importantly, successful probing critical nuclei help greatly in gaining insights into key aspects of diverse phase transitions.*

In this study, we employed the method to investigate the essential mechanisms during the nucleation of THF clathrate hydrates, specifically focusing on the critical-nuclei formation. Via the implementation of the method, the data analysis and the discussion of physical pictures from the experimental findings are further optimized; note that we found new results about the nucleation of THF hydrates significantly differed from the common pictures about the nucleation of clathrate hydrates in previous experiments and simulations. Indeed, during the development of science over time it is long observed that a new method is firstly invented and then implemented in various systems to tackle pending scientific questions or discover new mechanisms.

The discovery of larger critical nuclei sizes, such as around 30 nm for THF clathrate hydrate under median supercooling in comparison with the previous roughly estimated value of normal clathrate hydrates (smaller than 10 nm)³, is inherently significant in the field of clathrate phase transitions. It opens the door for further exploration of the structure and dynamics of critical nuclei of the THF (and another) clathrate hydrates.

Therefore, investigating the size of critical nuclei in THF clathrate hydrates adds a new dimension to our understanding of nucleation in different materials and provides insights into the unique factors governing clathrate hydrate formation. Probing the size of critical nuclei in THF clathrate hydrates not only addresses a critical knowledge gap specific to the clathrate hydrate formation but also contributes to a broader understanding of nucleation processes.

Moreover, the confirmation of the existence of critical nuclei and the determination of their size in THF

clathrate formation bear significant importance. The critical nucleus as the central concept and the control step for the clathrate hydrate formation has remained experimentally undetected. This study clearly reveals experimentally the key step during the THF clathrate hydrate nucleation, which can be employed to provide insights on the formation of the clathrate hydrates.

* The authors insist to claim that the nucleation of THF clathrate hydrates is representative of clathrate hydrates of gas, which is just incorrect and worlds apart. The nucleation of THF clathrate hydrates is indeed similar to ice as it is a bulk nucleation phenomenon, whereas, as pointed in the original review, for clathrate hydrates of gas, the process is severely mass transfer limited, which is not presented for THF. The vast majority of interest in clathrate hydrates is on gas, not THF. As such, there is limited translation of these learnings to system of actual interest. This does not question the scientific effort and gain from the work performed, but its impact and important is limited.

***Response:** We appreciate the valuable suggestions for identifying the distinctions between THF and gas hydrates. This suggestion has prompted us to have deeper consideration of the nucleation processes in both systems. Investigating the nucleation process of THF clathrate hydrates carries inherent significance⁴⁻⁶. THF's capability to form binary guest hydrates with gas hydrates, like methane and hydrogen, under relatively mild experimental conditions is noteworthy⁷. This inclusion of THF can reduce synthesis pressure by occupying a larger cavity, thereby stabilizing the material⁸. Therefore, a deeper understanding of the nucleation process of THF clathrate hydrates offers valuable insights for the storage and development of gas hydrates. Efforts have been put to study the nucleation process of THF hydrates. H. Conrad et al. studied the THF clathrate hydrate formation by x-ray Raman scattering measurements (Phys. Rev. Lett. 2009, 103, 218301). In this paper, the authors stated that the conclusions from THF hydrate have general impact for the research of hydrates via showing **“Both findings may have a great impact for the research of hydrates in general and for applications for gas storage, e.g., hydrogen”**. In addition, Zeng et al. studied the inhibition activities of two antifreeze proteins (AFPs) on the formation of THF clathrate hydrate, and based on which they tried to understand whether AFPs can inhibit the formation of natural gas hydrate (J. Am. Chem. Soc. 2006, 128, 2844-2850). They also draw a conclusion that **“Whatever the reason, inhibitors based on AFPs are likely to find utility in the prevention of unscheduled hydrate growth”**. And according to these impactful and groundbreaking literatures, we limit our findings and conclusion in the THF clathrate hydrates but not explicitly emphasize the possible difference of the nature gas clathrates from the THF hydrate.*

We also agree that there should be differences in the nucleation process between THF and gas clathrate

hydrates, primarily arising from the varied guest molecular transfer dynamics. During the clathrate nucleation, it involves the transferring of guest molecules within the solution to form the hydrate cages, and the transferring from the gas phase to the solution (for gas clathrates only). For the formation of critical nuclei (in several nanometers), it is directly determined by the local transferring of the dissolved guest molecule from the surrounding solution to form the clathrate nuclei, if the concentration of molecule in the solution is obviously different from that in the clathrate nuclei; while the formation of macroscopic clathrate hydrate crystal from the nuclei via apparent growth is relevant to the transferring of the guest molecules in global (long distance), e.g., from the gas phase to the solution. Note that current work concentrates on the pre-growth stage of hydrates, specifically the formation of critical nuclei from the dissolved guest molecule to form hydrate structures, but not the apparent growth of hydrate after the formation of nuclei via obvious transferring of molecules to form the macroscopic clathrate hydrate crystals. Additionally, it's essential to recognize that a nucleation process is a phenomenon occurring only when both thermodynamic and kinetic requirements are met. The nucleation of gas clathrate hydrate is mass transfer limited due to the low concentration of dissolved gas molecules in comparison with that in the forming clathrate nuclei, and it implies that the gas clathrate hydrate is hard to nucleate at least in kinetics, which is different from that of THF hydrate nucleation where the THF concentration in the solution is comparable to that in clathrates. The difference in affecting the formation of nanoscale critical nucleus should be much smaller than that in affecting the growth of nuclei to form the macroscopic clathrates. It is indeed an interesting and open question to investigate the extent of the difference between the THF and gas hydrates in forming critical nuclei further, which seems not so directly estimating as that in the macroscopic growth of the clathrate hydrates. We left it in next works.

The nucleation process of THF hydrate (THF-water binary structure) is notably more intricate than that of ice. The distinctive molecular components and crystal unit features between ice and THF hydrate underscore their substantial differences. Our findings reveal that the nucleation mode of THF clathrate hydrate and ice is not entirely identical.

We limited this current work in the THF hydrates and added discussions about the difference between the nucleation of THF and gas clathrate hydrates in the revised manuscript, as following:

Line 54: Differing from the high-pressure requirement needed for gas clathrate hydrates, the unique capability of THF hydrates to form clathrates independently and as binary guest hydrates with gases (such as methane and hydrogen) under relatively mild experimental conditions makes them particularly intriguing for clathrate hydrate research^{7,8}. Efforts have been put to study the nucleation process of THF hydrates, for example, via

the x-ray Raman scattering measurements⁴, and via regulating through antifreeze proteins (AFPs)⁶. The achieved conclusions were thought to be very helpful for the understanding of the gas clathrate hydrates.

Line 373: An intriguing and still open question is the comparison between the nucleation mechanism of the THF hydrates and the gas hydrates. A key distinction likely arises from the different mass transfer dynamics of guest molecules, involving the transfers of the dissolved guest molecules from the surrounding solution to form the clathrate hydrate nuclei, and the transfer of guest molecule from the gas phase into the solution for the apparent clathrate growth (specifically for gas clathrates). For the formation of nanoscale-size critical nuclei, it is mainly related to the local transferring kinetics of the dissolved guest molecule. In the case of gas clathrate, the low concentration of dissolved gas molecules in the solution results in a number ratio between gas guest and water molecules smaller than that in the formed clathrate, which may induce different effects from the process of forming THF hydrates from a miscible THF solution. It is worthwhile to investigate the possible effects in the formation of critical nuclei in the future.

Reviewer #2 (Remarks to the Author):

At a glance, the authors addressed some points the reviewer suggested previously. But, in the text, the discussion on the previously-reported nucleation mechanism of THF clathrate hydrate was insufficient. The authors just added the references in the text. They only discussed in the responses to reviewers. According to the reports by Machida et al., they observed the cluster in the THF aqueous solution by SEM before nucleation. They reported the size of clusters and the temperature dependence of number density of clusters. Probably the clusters in the literature would be equal to the critical nucleus the authors. Why the authors do not compare their results with the SEM observation results? The reviewer believes the addition of the quantitative discussion of the size makes this manuscript better. It should lead to further scientific understanding of the product of ΔT and L .

Response: Thanks for your comment, and we have discussed the results from the mentioned reference in the revision, as follows:

Line 64: We experimentally proved the formation of critical nucleus as the key step of the phase transition of tetrahydrofuran (THF) clathrate hydrate; and it is further showed that the spherical radius of the critical nucleus of the THF hydrates is about $300/\Delta T$ (nm), which is about a few times larger than that of ice in experiment¹ and that of gas clathrate hydrates in simulations³, but seems to be consistent with a recent (relevant) cryo-SEM experimental measurement, which reported the formation of about 10-30 nm nanoclusters in diameter of THF

clathrate hydrates before the appearance of THF clathrate crystals at the supercooling $\Delta T=20\text{ }^{\circ}\text{C}$ ⁹. The results indicate that the microscopic properties of the formed critical nuclei of THF hydrates and their surrounding solution differ from their macroscopic counterparts. This provides new insights about the detailed picture of the THF nucleation for further investigation.

Line 360: In other side, a recent publication reported the cryo-SEM observation of the formation of 10-30 nm nano-clusters of THF clathrate hydrate at a supercooling level of approximately 20 °C, before the appearance of THF clathrate crystals⁹, which is consistent with our results, as the critical nucleus diameter under 20 °C supercooling is approximately about $\frac{600\text{ nm}\cdot^{\circ}\text{C}}{20\text{ }(\Delta T)} = 30\text{ nm}$ (ΔL).

Reviewer #3 (Remarks to the Author):

In my original report I found the article to be of general interest and technically sound. I raised two questions which I feel have been satisfactorily addressed in the response.

"At a composition of 19 wt % the clathrate should be the only crystal phase seen, but in fact ice is also inferred from

DSC. Could this be due, for example, to THF adsorption on the nanoparticles? Can the quantity of ice be estimated?". The authors have responded to this point by clarifying the experimental method and possible origins of pure ice formation. This probably doesn't rule out THF adsorption on the nanoparticles, but in my view the conclusions are robust in any case.

"GO is soluble in water-THF mixtures, in which case could the experiment be conducted on dissolved rather than tethered nanoparticles?". The authors respond by explaining why solution studies might be more complicated than in pure water/ice, for example by GO folding, and have added to the main text.

Some minor comments/suggestions on the highlighted text.

Response: *We appreciate your comments in refining the manuscript.*

Title, I suggest: "Probing the critical nucleus size in tetrahydrofuran clathrate hydrates"

Response: *We have incorporated your recommendation and applied the suggested name in the revision.*

Line 92: "The remaining"

Response: *Thanks for your careful reading, and we followed your advice to refine our expressions in the revised manuscript.*

Line 264: "The dependence of the free energy barrier" (remove "In physics").

Response: Thanks. We followed your advice to remove the “In physics” in the revision.

Reviewer #4 (Remarks to the Author):

The effort taken by the authors to thoroughly respond to all reviewer comments is indeed commendable. I believe the manuscript still needs revision before it can be accepted for publication.

1. The critical nucleus size and number of cages reported in this study are significantly larger than what's reported in literature for clathrate hydrates. Authors should use their results to prove/disprove available MD simulation/experimental results on clathrate hydrate critical nucleus size. For example, Jacobson and Molinero (<https://pubs.acs.org/doi/pdf/10.1021/ja201403q>) report critical nucleus size of 3-4 nm. They also report subcooling vs. nucleus size from MD simulations. Does that agree with $R_c=300/dT$ found in this study? If not, why? It will add value to the manuscript if authors can compare their results with literature values and provide a discussion on why they think their results are different.

Response: Thanks for your suggestions. The size of critical nuclei is dependent on the temperature. In the mentioned reference of MD simulation on the size and crystallinity of critical clathrate nuclei from V. Molinero and coworkers, they gave the relation of $R_c = 73.2/\Delta T$ for the crystalline critical nuclei of methane clathrate and chose 36 mJ/m^2 as the surface tensions between liquid and sI methane clathrate crystal. Their value of $73.2 \text{ (nm } ^\circ\text{C)} = R_c \cdot \Delta T$ (in methane clathrate) is much smaller than that obtained in our experiment (for THF clathrate), about $300 \text{ nm } ^\circ\text{C}$. The disparity could be attributed to variations in the applied systems, with THF clathrate exhibiting an sII crystal structure, while methane clathrate adopts an sI crystal structure. As pointed by the reviewer 2, a recent publication reported the formation of 10-30 nm nano-clusters of THF clathrate hydrate at a supercooling level of approximately $20 \text{ } ^\circ\text{C}$ ¹⁰. Drawing from our results and analysis, we estimate that the stable critical nucleus diameter under $20 \text{ } ^\circ\text{C}$ supercooling is approximately about $600/20 (\Delta T) = 30 \text{ nm}$ (ΔL). Under this condition, as we use the GOs with a length size of 38 nm with an area of 361 nm^2 , considering the contact angle between critical nuclei and substrate (supposing it about 30 degrees), the critical nucleus could accommodate a maximum of $55448/1.731^3 = 10690$ unit cells. Therefore, it is possible that the nucleation of the THF hydrates may be different from that of methane hydrate, due to their different thermodynamical properties, and/or due to the difference of the nanoscale THF hydrate nuclei/surround solution from their macroscopic counterparts. More works on both the simulation and experiment sides, involving the hydrates of gas and that of THF, are needed to further clarify this deviation.

We have given the discussion about the published results in the revision, as follows:

Line 315: Especially, the size of critical nucleus, $R_c \sim \frac{300}{\Delta T}$ nm, inversely proportional to the supercooling temperature ΔT , is in well agreement with the expectation of CNT. Therefore, we actually present a nanometer ruler (nanosheets or nanocubes with a specific size) and detect the nucleation signal on it. When the size of critical nucleus (dependent on ΔT) matches with that of the nanometer ruler, the clathrate forms thus the size of critical nucleus can be recorded. It is worth mentioning that current work about THF clathrate nucleation gives the size of critical nucleus of THF hydrates being about a few larger than that of the ice nucleus¹, which is reported for first time by quantitative analysis.

Line 356: Thus, we roughly estimated the macroscopic value of the THF hydrates $\frac{\gamma}{|\Delta S|} \lesssim 40$ nm °C. This estimate is consistent with the recent results about the methane clathrate by MD simulations³, where $R_c \approx 73.2/\Delta T$ (nm), i.e., $\frac{\gamma}{|\Delta S|} \approx 36.6$ nm °C, by choosing 36 mJ/m² as the surface tensions between liquid and solid methane clathrate crystal, but much smaller than the obtained microscopic value obtained from our current experiment, about 150 nm °C. In other side, a recent publication reported the cryo-SEM observation of the formation of 10-30 nm nano-clusters of THF clathrate hydrate at a supercooling level of approximately 20 °C, before the appearance of THF clathrate crystals⁹, which is consistent with our results, as the critical nucleus diameter under 20 °C supercooling is approximately about $\frac{600 \text{ nm } ^\circ\text{C}}{20 (\Delta T)} = 30$ nm (ΔL). Therefore, besides the possible difference in the estimated values of the melting entropy and surface tension of the THF clathrates from that of gas clathrates, as well as the possible deviation between simulations and experiments, the found large critical nucleus of THF clathrates in the current experiments may indicate the difference of the nanoscale THF hydrate nuclei/surrounding solution from their macroscopic counterparts. The conclusion, which still needs further verification by more simulations and experiments, is consistent with the findings in MD simulations about the occurrence of some pre-nucleation steps which might lead different structure and character of surrounding solution of critical nucleus from the macroscopic solution¹¹.

2. Regarding thermal conductivity of nanoparticles, please see following works on size effects:

<https://pubs.acs.org/doi/10.1021/nl1045395>

<https://journals.aps.org/prb/abstract/10.1103/PhysRevB.61.2651>

In their analysis, authors assumed same thermal conductivity for different sized particles. In fact, thermal conductivity of smaller nanoparticles in their work can be 1-2 order of magnitude lower than bulk value. The thermal analysis is not straight forward as authors assumed.

***Response:** Thanks for the suggestions. We find that the mentioned two reference discussing the size effect of nanocrystalline silicon on their thermal conduction, noting that the thermal conductivity of nano-sized silicon is lower than that of its bulk counterpart^{12,13}. This is attributed to the nano-structures, which possess large surface-volume ratios, leading to heightened surface phonon scattering¹⁴.*

However, these principles may not be directly applicable to metallic materials like gold and silver. For the gold nanoparticles, the thermal conduction is often used as 315 W/(m·K)^{15,16}, which is similar as the bulk gold material. For the silver nanoparticles, the thermal conduction is often used as 429 W/(m·K)^{17,18}, which is similar as the bulk silver material. In addition, the graphene oxide experiences a decrease in thermal conduction in nano-size, with a reported value of 18 W/(m·K)¹⁹, resulting in the temperature difference of $\Delta T_{GO} = 0.014$ °C. Despite this decrease, the impact remains limited, exerting less influence on local temperature during nucleation. We added the discussion in the revised manuscript, as followed:

*Line 256: Noted that the other factors, such as the different heat conductance for nanoparticles of various size, and the dynamics of molecules neighboring nanoparticles (**Methods**), are not possible to contribute to the observed transition occurring at the specific value of $\Delta T \cdot L$ under all these different experimental conditions, especially the delay time measurements with fixing both the size of nanoparticles and supercooling.*

Line 479: at the nanoscale, the calculated temperature difference between the top and bottom surfaces of the nanoparticles (including melt nanocubes and GO nanosheets) is within 0.03 °C. This marginal temperature difference has no impact on the statistical analysis of nucleation temperatures, consequently, does not influence the conclusions drawn from the study.

In addition, in our work, the transition of nucleation capability of nanoparticles (nanosheets of GO, nanocubes of Au/Ag) occurs at a supercooling-dependent size, not a specific size, and the transition is abrupt (not from the bulk to nanometer, but between a few nanometers in size, even in the same size but varying supercooling less 1°C), therefore, the possible large difference between some bulk materials and its nanoscale counterparts

cannot be related to our findings.

Reference:

- 1 Bai, G., Gao, D., Liu, Z., Zhou, X. & Wang, J. Probing the critical nucleus size for ice formation with graphene oxide nanosheets. *Nature* **576**, 437-441, doi:10.1038/s41586-019-1827-6 (2019).
- 2 Wilson, P. W. & Haymet, A. D. J. Hydrate formation and re-formation in nucleating THF/water mixtures show no evidence to support a “memory” effect. *Chemical Engineering Journal* **161**, 146-150, doi:10.1016/j.cej.2010.04.047 (2010).
- 3 Jacobson, L. C. & Molinero, V. Can amorphous nuclei grow crystalline clathrates? The size and crystallinity of critical clathrate nuclei. *J Am Chem Soc* **133**, 6458-6463, doi:10.1021/ja201403q (2011).
- 4 Conrad, H. *et al.* Tetrahydrofuran clathrate hydrate formation. *Phys Rev Lett* **103**, 218301, doi:10.1103/PhysRevLett.103.218301 (2009).
- 5 Sugahara, T. *et al.* Increasing hydrogen storage capacity using tetrahydrofuran. *J Am Chem Soc* **131**, 14616-14617, doi:10.1021/ja905819z (2009).
- 6 Zeng, H., Wilson, L. D., Walker, V. K. & Ripmeester, J. A. Effect of antifreeze proteins on the nucleation, growth, and the memory effect during tetrahydrofuran clathrate hydrate formation. *J Am Chem Soc* **128**, 2844-2850, doi:10.1021/ja0548182 (2006).
- 7 Lee, H. *et al.* Tuning clathrate hydrates for hydrogen storage. *Nature* **434**, 743-746, doi:10.1038/nature03457 (2005).
- 8 Florusse, L. J. *et al.* Stable low-pressure hydrogen clusters stored in a binary clathrate hydrate. *Science* **306**, 469-471, doi:10.1126/science.1102076 (2004).

- 9 Machida, H., Sugahara, T. & Hirasawa, I. Supercooling suppression in the tetrahydrofuran clathrate hydrate formation. *CrystEngComm* **24**, 6730-6738, doi:10.1039/d2ce00645f (2022).
- 10 Jin, L., Shi, Y., Allen, F. I., Chen, L. Q. & Wu, J. Probing the Critical Nucleus Size in the Metal-Insulator Phase Transition of VO₂. *Phys Rev Lett* **129**, 245701, doi:10.1103/PhysRevLett.129.245701 (2022).
- 11 Sosso, G. C. *et al.* Crystal Nucleation in Liquids: Open Questions and Future Challenges in Molecular Dynamics Simulations. *Chem Rev* **116**, 7078-7116, doi:10.1021/acs.chemrev.5b00744 (2016).
- 12 Volz, S. G. & Chen, G. Molecular-dynamics simulation of thermal conductivity of silicon crystals. *Physical Review B* **61**, 2651-2656, doi:10.1103/PhysRevB.61.2651 (2000).
- 13 Wang, Z., Alaniz, J. E., Jang, W., Garay, J. E. & Dames, C. Thermal conductivity of nanocrystalline silicon: importance of grain size and frequency-dependent mean free paths. *Nano Lett* **11**, 2206-2213, doi:10.1021/nl1045395 (2011).
- 14 Li, D. *et al.* Thermal conductivity of individual silicon nanowires. *Applied Physics Letters* **83**, 2934-2936, doi:10.1063/1.1616981 (2003).
- 15 Rashidi, S., Karimi, N., Mahian, O. & Abolfazli Esfahani, J. A concise review on the role of nanoparticles upon the productivity of solar desalination systems. *Journal of Thermal Analysis and Calorimetry* **135**, 1145-1159, doi:10.1007/s10973-018-7500-8 (2018).
- 16 Elango, T., Kannan, A. & Kalidasa Murugavel, K. Performance study on single basin single slope solar still with different water nanofluids. *Desalination* **360**, 45-51, doi:10.1016/j.desal.2015.01.004 (2015).
- 17 Seyhan, M., Altan, C. L., Gurten, B. & Bucak, S. The effect of functionalized silver nanoparticles over the thermal conductivity of base fluids. *AIP Advances* **7**, doi:10.1063/1.4979554 (2017).
- 18 Simpson, S., Schelfhout, A., Golden, C. & Vafaei, S. Nanofluid Thermal Conductivity and Effective Parameters. *Applied Sciences* **9**, 87, doi:10.3390/app9010087 (2018).

- 19 Mahanta, N. K. & Abramson, A. R. Thermal conductivity of graphene and graphene oxide nanoplatelets. 1-6, doi:10.1109/itherm.2012.6231405 (2012).

REVIEWERS' COMMENTS

Reviewer #2 (Remarks to the Author):

The authors addressed my comments and suggestions properly. Though an ambiguous point remains in the scientific meanings of the product of ΔT and L , the reviewer believes that this manuscript will be an important step to elucidate the novel crystallization mechanism. Therefore, the reviewer recommends the publication in Nature Communications.

Reviewer #4 (Remarks to the Author):

As before, authors have done a good job in answering my comments. Their efforts are indeed commendable. I have some additional suggestions which may be useful for future work.

1. On comparison of critical nucleus from MD simulations vs. current work:

I feel the authors limited their response to the paper I suggested in my previous review and didn't look for more. There are many other MD studies that have looked into hydrate nucleation at relatively moderate conditions as well as in presence of a surface. For example:

<https://pubs.acs.org/doi/10.1021/acs.jpcc.5b10293>

<https://www.pnas.org/doi/10.1073/pnas.1906502116>

<https://pubs.acs.org/doi/10.1021/ja400521e>

<https://www.nature.com/articles/srep12747>

The first reference above on THF hydrate growth may be specially relevant.

2. Regarding size effects on thermal conductivity:

I do not agree with the reported thermal conductivity values for gold and silver nanoparticle being same as that of bulk. It depends on the size of the particle, especially when particle size is of the same order of magnitude as phonon/electron mean free path. See this reference:

<https://link.springer.com/book/10.1007/978-3-030-45039-7>

It could be that nanoparticles used in those studies were large enough that the size effects did not come into picture.

For graphene oxide nanosheets, situation is even more complicated. Anisotropy (in-plane vs. out-of-plane) and surface defects and different oxygen concentration at surface have drastic effect on its thermal conductivity. See for example:

<https://www.nature.com/articles/srep03909>

<https://onlinelibrary.wiley.com/doi/abs/10.1002/adfm.201501429>

In any case, these comments are more relevant for future work.

I believe after some minor changes in manuscript to reflect above comments, manuscript can be accepted for publication.

Reviewer #2 (Remarks to the Author):

The authors addressed my comments and suggestions properly. Though an ambiguous point remains in the scientific meanings of the product of ΔT and L, the reviewer believes that this manuscript will be an important step to elucidate the novel crystallization mechanism.

Therefore, the reviewer recommends the publication in Nature Communications.

Response: We appreciate the reviewer's comments in refining the manuscript.

Reviewer #4 (Remarks to the Author):

As before, authors have done a good job in answering my comments. Their efforts are indeed commendable. I have some additional suggestions which may be useful for future work.

1. On comparison of critical nucleus from MD simulations vs. current work:

I feel the authors limited their response to the paper I suggested in my previous review and didn't look for more. There are many other MD studies that have looked into hydrate nucleation at relatively moderate conditions as well as in presence of a surface.

For example:

<https://pubs.acs.org/doi/10.1021/acs.jpcc.5b10293>

<https://www.pnas.org/doi/10.1073/pnas.1906502116>

<https://pubs.acs.org/doi/10.1021/ja400521e>

<https://www.nature.com/articles/srep12747>

The first reference above on THF hydrate growth may be especially relevant.

Response: Thanks for your suggestions. After carefully reading the mentioned references. The first article shows that a THF molecule trapped in an open small cage needs to cross one or two free energy barriers to escape from the surface region, and thus the crystal growth of THF hydrate is much slower than that of ice.

Both of the third and fourth articles show the effect of surface properties on the hydrate formation. These references will be significant for the future study of the interaction between surface and hydrate nuclei, and the kinetics of hydrate critical nuclei dynamics. We add the discussion in the revised manuscript, as following:

Line 223: Further investigation into the static structure and dynamic behavior of hydrate critical nuclei will yield additional experimental evidence for understanding the interactions between the nuclei and surrounding solutions at the nanoscale¹⁻⁴.

The second article simulated the size of critical methane hydrate nuclei size as 300 mutually coordinated guest (methane molecules), which is still smaller than our results. This article has been incorporated into the existing discussion in our revised manuscript.

2. Regarding size effects on thermal conductivity:

I do not agree with the reported thermal conductivity values for gold and silver nanoparticle being same as that of bulk. It depends on the size of the particle, especially when particle size is of the same order of magnitude as phonon/electron mean free path. See this reference:

<https://link.springer.com/book/10.1007/978-3-030-45039-7>

It could be that nanoparticles used in those studies were large enough that the size effects did not come into picture.

For graphene oxide nanosheets, situation is even more complicated. Anisotropy (in-plane vs. out-of-plane) and surface defects and different oxygen concentration at surface have drastic effect on its thermal conductivity. See for example:

<https://www.nature.com/articles/srep03909>

<https://onlinelibrary.wiley.com/doi/abs/10.1002/adfm.201501429>

In any case, these comments are more relevant for future work.

I believe after some minor changes in manuscript to reflect above comments, manuscript can be accepted for publication.

Response: Thanks for your suggestions. We limited the description of the relationship between the nanoparticle size and the heat conductivity in the revised manuscript, as following:

Line 295: Moreover, for the used nanoparticles (including melt nanocube with size large than 45 nm, and GO nanosheets with size larger than 30 nm) in this study, the calculated temperature difference between the top and bottom surfaces of the nanoparticles is within 0.03 °C.

Reference:

- 1 Yagasaki, T., Matsumoto, M. & Tanaka, H. Mechanism of Slow Crystal Growth of Tetrahydrofuran Clathrate Hydrate. *The Journal of Physical Chemistry C* **120**, 3305-3313, doi:10.1021/acs.jpcc.5b10293 (2016).
- 2 Pirzadeh, P. & Kusalik, P. G. Molecular insights into clathrate hydrate nucleation at an ice-solution interface. *J Am Chem Soc* **135**, 7278-7287, doi:10.1021/ja400521e (2013).
- 3 Bai, D., Chen, G., Zhang, X., Sum, A. K. & Wang, W. How Properties of Solid Surfaces Modulate the Nucleation of Gas Hydrate. *Scientific reports* **5**, 12747, doi:10.1038/srep12747 (2015).
- 4 Sosso, G. C. *et al.* Crystal Nucleation in Liquids: Open Questions and Future Challenges in Molecular Dynamics Simulations. *Chem Rev* **116**, 7078-7116, doi:10.1021/acs.chemrev.5b00744 (2016).